

# Atmospheric Processing of Iron in Mineral and Combustion Aerosols: Development of an Intermediate-Complexity Mechanism Suitable for Earth System Models.

Rachel A. Scanza[1,2], Natalie M. Mahowald[1], Carlos Perez Garcia-Pando[3], Clifton Buck[4], Alex Baker[5],
Douglas S. Hamilton[1]

[1]Department of Earth and Atmospheric Sciences, Cornell University, Ithaca, New York, USA
[2]Atmospheric Sciences and Global Change Division, Pacific Northwest National Laboratory, Richland, Washington, USA
[3]Earth Sciences Department, Barcelona Supercomputing Center, Barcelona, Spain
[4]Department of Marine Sciences, University of Georgia, Athens, Georgia, USA
[5]School of Environmental Sciences, University of East Anglia, Norwich, UK

*Correspondence to*: Rachel A. Scanza (Rachel.scanza@pnnl.gov)

**Abstract.** Atmospheric processing of iron in dust and combustion aerosols is simulated using an intermediate-complexity soluble iron mechanism designed for Earth system models. The solubilization mechanism includes both a dependence on aerosol water pH and in-cloud oxalic acid. The simulations of size resolved total, soluble and fractional iron solubility indicate that this mechanism captures many but not all of the features seen from cruise observations of labile iron. The primary objective was to determine the extent to which our solubility scheme could adequately match observations of fractional iron solubility. We define a semi-quantitative metric as the model mean at points with observations divided by the observational mean (MMO); fractional iron solubility MMO is 0.8, indicating that while the model is not capturing all of the observational variability, it is within range of the observational mean. Several sensitivity studies are performed to ascertain the degree of complexity needed to match observations; including the oxalic acid enhancement is necessary while different parameterizations for calculating model oxalate concentrations are less important. The percent change in soluble iron deposition between the reference case and the simulation with acidic processing alone is 63.8%, which is consistent with previous studies. Upon deposition to global oceans, global mean combustion iron solubility to total fractional iron solubility is 8.2%; however, the contribution of fractional iron solubility from combustion sources to ocean basins below 15°S is approximately 50%. We conclude that in many remote ocean regions, sources of iron from combustion and dust aerosols are equally important. Our estimates of changes in deposition of soluble iron to the ocean since preindustrial suggest roughly a doubling due to a combination of higher dust and combustion iron emissions along with more efficient atmospheric processing.

## 1 Introduction

Nearly all ocean productivity relies on enzymatic iron for photosynthesis, respiration and nitrogen fixation. The iron biogeochemical cycle is therefore an important modulator of the oceans ability to uptake atmospheric $CO_2$ Since



approximately 30% of atmospheric $CO_2$ emissions are absorbed by the oceans, (Seiler and Crutzen, 1980;Broecker et al., 1979;Siegenthaler and Sarmiento, 1993) changes to the iron cycle may induce a potentially large negative feedback on the climate system (Martin and Fitzwater, 1988;Martin, 1990;De Baar et al., 1995;Jickells and Spokes, 2001;Jickells et al., 2005).

5     Many remote ocean regions are characterized by high-nutrient low-chlorophyll (HLNC) conditions; in the equatorial Pacific and southern ocean basins, ocean primary productivity (OPP) is iron limited (Martin and Fitzwater, 1988;Boyd et al., 2000). A significant source of new iron (in contrast to re-circulated iron from upwelling, estuary runoff or fluxes from continental margins) to these regions is atmospheric deposition of desert dust (Jickells et al., 2005;Fung et al., 2000), however, iron in dust is typically insoluble and not considered to be bioavailable for OPP (Mahowald et al., 2005;Johnson, 2001;Jickells and Spokes, 2001). Although combustion iron is a much smaller source of new iron, it is 10 considerably more soluble than dust iron (Guieu et al., 2005;Chuang et al., 2005), potentially contributing up to 50% of bioavailable iron in remote HNLC regions (Luo et al., 2008;Mahowald et al., 2009;Winton et al., 2015;Ito, 2015). Many definitions exist for bioavailable iron although how organisms utilize iron is not well understood (Jickells et al., 2005). Bioavailable iron is typically considered to be dissolved iron in the ferrous (Fe(II)) oxidation state; however, ocean 15 organisms have been observed to utilize iron in different forms (Barbeau et al., 1996). In this study, bioavailable iron is defined as labile iron, or dissolved iron in either the ferrous or ferric oxidation states.

Acidic processing of mineral dust and combustion aerosols during atmospheric transport is one potential mechanism for iron dissolution upon deposition, and many studies have observed increased iron liberation from insoluble iron oxides with decreasing pH (Duce and Tindale, 1991;Zhu et al., 1997;Zhuang et al., 1992;Jickells and Spokes, 20 2001;Desboeufs et al., 2001;Meskhidze et al., 2003). However, acidic processing is likely to work in conjunction with other physical and chemical processes such as photochemical reduction and organic ligand processing. Iron solubility has also been correlated with particle size with increasing fractional iron solubility as mineral dust concentrations decrease (Baker and Croot, 2010;Chen and Siefert, 2004;Baker and Jickells, 2006). This relates to the preferential settling of larger particles during transport along with a higher surface area to volume ratio for smaller particles. Because smaller particles are 25 associated with longer transport times, their probability of undergoing multiple cycles of evaporation and condensation in the atmosphere is increased. Oxalate, the oxidation product of oxalic acid, can act as an organic ligand; this has been observed to enhance iron dissolution when complexation with aerosols occurs in aqueous solutions under moderately acidic conditions (e.g. within clouds) (Cornell and Schindler, 1987;Xu and Gao, 2008;Solmon et al., 2009;Paris et al., 2011). The formation of oxalate in the atmosphere is complicated (Myriokefalitakis et al., 2011) but may increase soluble iron concentrations over 30 75% in addition to acidic processing alone (Johnson and Meskhidze, 2013;Myriokefalitakis et al., 2015;Ito, 2015). Previous studies focused on observations have shown a very strong inverse relationship between total iron and solubility (Sholkovitz et al., 2012), where differences in methods of measuring soluble iron appear to be much less important than previously thought (Baker et al., 2014).



The type of iron in dust is dependent on mineralogy and surface area. For example, iron associated with hematite and goethite is tightly bound as iron (hydr)oxides and its dissolution even under very acidic conditions is slow. Nanoparticles of hematite and ferrihydrite have been observed to coat clay minerals illite and smectite (Shi et al., 2012;Ito and Xu, 2014); the surface area to volume ratio in nanoparticles (<0.1 μm) is sufficiently high to facilitate rapid dissolution

of Fe at low pH. The clay minerals illite, smectite, feldspar and kaolinite also include Fe as substitutions within aluminosillicate mineral lattices and may be appreciable sources of soluble Fe (Journet et al., 2008). Different levels of model complexity have been employed to simulate atmospheric iron processing from very simple models including a first order rate constant applied to a constant 3.5% iron in dust (Hand et al., 2004;Luo et al., 2008) to more complex schemes allowing different types of acidic species to interact with mineral dust and combustion aerosols during transport and

applying mineral-specific dissolution rates (Meskhidze et al., 2005;Johnson and Meskhidze, 2013;Ito and Xu, 2014;Myriokefalitakis et al., 2015;Ito, 2015).

In this study, our goal is to develop an iron dissolution scheme of intermediate complexity that can be used in Earth system models. Recent work has emphasized the importance of variable dissolution rates based on pH and chemical composition(Meskhidze et al., 2005;Johnson and Meskhidze, 2013;Ito and Xu, 2014;Myriokefalitakis et al., 2015;Ito, 2015)

and this requires the advection of multiple chemical species and calculations of chemical equilibrium. We include these complex mechanisms but use simplified parameterizations based on standard aerosol species available in most Earth system models. The type and amount of iron at emission is determined from mineral maps and brittle fragmentation theory (Claquin et al., 1999;Scanza et al., 2015;Kok, 2011). We simulate four types of iron (readily-released Fe, medium-soluble Fe, slow-soluble Fe and combustion Fe) (Ito and Xu, 2014;Ito, 2015). An acid processing scheme is employed based on atmospheric

concentrations of sulfate along with an in-cloud oxalate mechanism derived from laboratory data from (Paris et al., 2011) (Johnson and Meskhidze, 2013;Myriokefalitakis et al., 2015;Ito, 2015). Our method performs as well as more computationally intensive methods in matching available observations of soluble iron. Additionally, the parameterizations and simplifications incorporated in our scheme renders it computationally efficient, and thus allows for multi-decadal to centennial simulations.

The study is organized as follows: Section 2 describes the climate and dust model we use along with the details of our iron processing scheme. Section 3 compares the results of the model simulations to available observations of soluble iron using a set of metrics established to evaluate model accuracy. Additionally, results from six sensitivity studies are discussed in order to quantify which model parameterizations are required to best match observations. Section 4 presents the results from four preindustrial simulations. Section 5 includes a discussion and comparison to previous modelling studies. The

final section describes where future work should be directed.

## 2 Methods

The Community Earth System Model (CESM) coordinated by the National Center for Atmospheric Research (NCAR) is a global Earth system model used for past, present and future climate simulations. In this study we use the CESM version



1.0.5 with a bulk aerosol model, the Community Atmosphere Model version 4 (CAM4) (Neale et al., 2013). Previous versions of the model have been modified to allow for improved treatment of dust and mineral speciation (Albani et al., 2014;Scanza et al., 2015;Zhang et al., 2015), and here we further modify CAM4 in order to simulate the emission, transport and processing, and deposition of total and soluble iron in desert dust and combustion aerosols.

## 2.1 Desert dust model

CAM4 is used to simulate the emission, transport and deposition of bulk aerosols in the CESM. Major aerosol species include dust, sea salt, black carbon, organic carbon (OC), and sulfate, and are prescribed as external mixtures. Simulations are performed at a horizontal resolution of 1.9° x 2.5° with 56 vertical levels that span the pressure at the surface to around 2 hPa (Computational and Information Systems Laboratory. 2012. Yellowstone: IBM iDataPlex System (Climate Simulation Laboratory). Boulder, CO: National Center for Atmospheric Research. http://n2t.net/ark:/85065/d7wd3xhc). Meteorology is driven by NASA's Goddard Earth Observation System (GEOS-5) (Suarez et al., 2008;Hurrell et al., 2013;Lamarque et al., 2012). The model is run from 2005 through 2011 assuming present day climate conditions; we use the last five (2007-2011) for analysis.

The desert dust model in CAM4 is a modified version of the Dust Entrainment and Deposition model (DEAD) (Zender et al., 2003), and is exactly as described and used in (Albani et al., 2014) and (Scanza et al., 2015). Briefly, the emission of dust is computed with an analytic trimodal lognormal probability density function from three source modes to four transport bins. Bin widths are prescribed at 0.1-1.0, 1.0-2.5, 2.5-5.0, and 5.0-10.0 μm and have fixed lognormal sub-bin distributions. The size distribution across dust bins was modified from the release version of the model to follow the brittle fragmentation theory of dust (Kok, 2011) with particle mass fractions of 0.011, 0.087, 0.272 and 0.625 in bins 1-4, respectively (Albani et al., 2014). Dry removal of dust aerosols involves parameterizations for gravitational settling and turbulent mix out and wet removal includes in-cloud and below-cloud scavenging (Rasch et al., 2000;Zender et al., 2003;Mahowald et al., 2006). The in-cloud aerosol removal rate is proportional to the fraction of cloud water that precipitates; some of the falling precipitation is allowed to re-evaporate (Rasch and Kristjánsson, 1998;Rasch et al., 2000). CAM4 allows for multiple cycles of condensation and evaporation in order to match observational estimates of approximately three in-cloud cycles for aerosols (Lelieveld et al., 1998;Crutzen and Zimmermann, 1991). Modifications from the release version include changes to the scavenging coefficients (from 0.1 to 0.3 for bins 3 (2.5-5.0μm) and 4 (5.0-10.0μm)), to the prescribed particle solubility from 0.15 to 0.3 across all size bins, and to the threshold for dust generation of the leaf area index from 0.15 up to 0.3 (Lancaster and Baas, 1998;Mahowald et al., 2006;Okin, 2008;Albani et al., 2014;Scanza et al., 2015).




## 2.2 Mineralogy

Mineralogy maps are derived from (Claquin et al., 1999) and are gridded using FAO/UNESCO WGB84 at 5' x 5' arc minutes with soil legend 184 from FAO/UNESCO Soil Map of the World (1976; File Identifier: f7ccd330-bdce-11db-a0f6-185-000d939bc5d8) (Batjes, 1997). The conversion from soil to aerosol mineralogy for each transport bin follows the brittle

fragmentation theory of dust (Kok, 2011) and is described in detail in (Scanza et al., 2015). The mineralogy maps are based on measurements performed following wet sieving, a process that destroys the mineral aggregates found in unperturbed parent soils. Brittle fragmentation theory reconstructs the mineral aggregation found at emission, allowing for more accurate reproduction of size-resolved dust and iron concentrations (Perlwitz et al., 2015a, b;Pérez García−Pando et al., 2016). We determine iron concentrations at emission from the derived aerosol mineralogy; therefore, we eliminate the need to explicitly

advect individual mineral tracers with the exception of calcite in order to parameterize the buffer effect of mineral dust on aerosol pH (Böke et al., 1999). Evaluation of the mineral distributions against observations of dust plume events showed some skill in the simulation of concentrations of minerals (Scanza et al., 2015); in addition, the resulting elemental distributions of iron, aluminium and calcium were improved through the use of the soil map (Zhang et al., 2015).

## 2.3 Determination of Iron

Iron in mineral dust is found in several forms and the solubility depends on the type of chemical bonding (or compositional form) (i.e. iron oxides vs. aluminosillicate inclusions) (Lafon et al., 2004;Journet et al., 2008) and on the particle size (Baker and Jickells, 2006). The type of iron is mineral dependent: aluminosillicates contain Fe inclusions as well as surface

coatings of nanohematite and ferrihydrite (Shi et al., 2009;Shi et al., 2012;Ito and Xu, 2014). Fe associated with hematite and goethite is tightly bound to oxygen in iron oxides/hydroxides, is typically larger in size and less soluble. Following previous studies, we define the Fe concentration in mineral dust as follows: 57.5% Fe in hematite, 11% in smectite, 4% in illite, 0.24% in kaolinite and 0.34% in feldspar (Journet et al., 2008;Ito and Xu, 2014). Three types of dust Fe are simulated in the model: readily-released iron ($Fe_{RR}$), medium-soluble iron ($Fe_{med}$) and slow-soluble or refractory iron ($Fe_{slow}$). The

$Fe_{RR}$ is assumed here as 2.7% of the 4% Fe in illite as ferrihydrite, 5% of the 11% Fe in smectite as nanohematite, 4.3% of the 0.24% Fe in kaolinite and 3% of the 0.34% Fe in feldspar. $Fe_{med}$ is defined as the remaining 97.3% and 95% Fe in illite and smectite, respectively. $Fe_{slow}$ is defined as the remaining 95.7% Fe in kaolinite, 97% in feldspar and 100% of the 57.5% Fe in hematite (Table S1). Separate tracers are defined for total medium soluble iron ($Fet_{med}$) and slow soluble iron ($Fet_{slow}$) in each of the four CAM4 size bins along with separate tracers for medium and slow soluble iron ($Fes_{med}$ and $Fes_{slow}$). In the

finest particle size bin (0.1-1.0 μm), as an approximation, we assume that $Fe_{RR}$ has already been solubilized and is added as $Fes_{med}$ at emission (Ito and Xu, 2014). In bins 2-4, the $Fe_{RR}$ associated with kaolinite and feldspar is also assumed to be solubilized (Ito and Xu, 2014) along with one quarter of the $Fe_{RR}$ associated with illite and smectite and is prescribed to




$Fes_{med}$ at emission. The distribution of iron from the silt and clay fractions in the soils follows the brittle fragmentation theory of dust emission (Kok, 2011), and is described in more detail in Scanza et al. (2015) in Table 2a.

Total iron in combustion aerosols ($Fet_{comb}$) is prescribed and industrial sources of iron are emitted following (Luo et al., 2008). Biomass burning is assumed to have a ratio of 0.02 g Fe/g BC in the fine mode and 1.4 g Fe/g BC in the coarse

mode (Luo et al., 2008). $Fet_{comb}$ is partitioned in the model transport bins as follows: fine mode $Fet_{comb}$ solely comprises the first bin and the coarse mode combustion iron is apportioned into bins 2-4 at 20, 30 and 50%, respectively. Separate tracers account for $Fet_{comb}$ and $Fes_{comb}$ with 4% of $Fet_{comb}$ assumed already soluble and prescribed to $Fes_{comb}$.

**2.4 Iron Dissolution**

Fe dissolution in mineral dust involves complex atmospheric chemical and physical processes. The major processes, i.e. processes that can be observed in laboratory settings, are related to the initial mineral composition, atmospheric temperature and acidity, insolation and concentration of organic acids. In this study, the amount of Fe solubilized in dust is assumed to be irreversible and is calculated using a simplified iron dissolution scheme for three types of iron in dust, $Fe_{RR}$, $Fe_{med}$ and $Fe_{slow}$. However, we only explicitly simulate $Fe_{med}$ and $Fe_{slow}$ both for computational efficiency and because we also assume

that a portion of $Fe_{RR}$ has already been solubilized at emission.

The CAM4 sulfur chemistry model is not configured to interact with dust; thus the rate of dissolution is only explicitly dependent on temperature and is simplified to a first order dissolution rate following Eq. (1),

$$RFe_i = K_i(T) \times a(H^+)^{m_i} \times f(\nabla G_r) \times A_i \times MW_i$$

(1)

$$\frac{d}{dt}[Fe_{soluble}] = RFe_i \times [Fe_{insoluble}]$$

where $i$ represents either medium or slow soluble Fe, $K_i(T)$ in units of (moles m$^{-2}$ s$^{-1}$) is the temperature dependent rate coefficient, $a(H^+)$ is the proton concentration with an empirical reaction order $m_i$, $f(\nabla G_r)$ accounts for the change in the dissolution rate with variation from equilibrium (and equals 1 for simplicity (Luo et al., 2008)), $A_i$ in units of (m$^2$ g$^{-1}$) of mineral$_i$ and $MW_i$ is the molecular weight in units of (g mol$^{-1}$) for mineral$_i$. The pH dependence is parameterized from the

concentration of calcite and sulfate at each atmospheric grid box and at each timestep. When [calcite]$_{i,j,k}$ > [sulfate]$_{i,j,k}$, pH is set to 7.5; when model sulfate concentration is greater, pH is set to 2 (Journet et al., 2008;Luo et al., 2008). $Fe_{med}$ is comprised of the Fe in illite and smectite; because the mineral abundance of illite is twice that of smectite, we use $K_{illite}(T)$ and $MW_{illite}$ for the $Fe_{med}$ dissolution rate as an additional simplification. The dissolution of Fe in hematite proceeds in three stages (Meskhidze et al., 2005) and we use the fastest of the three stages for $K_{hem}(T)$ following (Ito and Xu, 2014) (Figure 1).

The initial concentration of $Fes_{comb}$ is assumed to be 4% of $Fet_{comb}$ (Chuang et al., 2005;Luo et al., 2008) in each size bin and we use the $Fe_{med}$ dissolution rate for the remaining combustion iron.



The impact of including oxalate in an iron dissolution scheme can increase the soluble Fe fraction considerably (Paris et al., 2011;Myriokefalitakis et al., 2011;Johnson and Meskhidze, 2013;Myriokefalitakis et al., 2015;Ito, 2015). Because the formation of atmospheric oxalate is complex, we develop a simple scheme to estimate oxalate concentrations in the model at grid cells within clouds. The concentration of oxalate in µmols L$^{-1}$ is calculated following Eq. (2):

$$[C_2O_4^{2-}]_{i,j,k} = 15 \times \frac{[OC_{i,j,k}]+[SOA_{i,j,k}]}{max[OC+SOA]} \tag{2}$$

This simple approximation is liable to add a source of uncertainty into our calculations; however the spatial distribution of oxalate at the surface (Figure 2a) is comparable to Figure 2a,c in (Myriokefalitakis et al., 2011), which calculates oxalate

concentrations using a full complexity chemical mechanism. Previous studies use sulfate as a proxy for calculating model oxalate concentrations (Yu et al., 2005;Johnson and Meskhidze, 2013), and we conduct a sensitivity study to determine the relative importance of this assumption. Because the sources of OC and secondary organic aerosols (SOA) are different than the sources of sulfate, the distribution is different between these two 'proxies' for oxalate; the OC+SOA in our model appears to be a better proxy. There tends to be more oxalate in the model simulations in tropical regions (Figure 2 from

(Myriokefalitakis et al., 2011)) which is better captured in our model simulations using the OC+SOA versus the sulfate proxy for modeled oxalate concentrations.

The oxalate dependent reaction rate is then added to the first order dissolution rate (Equation 1) as follows in Eq. (3):

$$RFe_i = K_i(T) \times a(H^+)^{m_i} \times f(\nabla G_r) \times A_i \times MW_i + K_{i,oxalate} \tag{3}$$

where $K_{i,oxalate}$ is

$$K_{i,oxalate} = a_i \times [C_2O_4^{2-}] + b_i$$

Coefficients $a_i$ for illite and hematite are determined from the slope of the best fit of the data in Table 4 in (Paris et al., 2011)

and $b_i$ for illite and hematite correspond to the intercept of the best fit equations. For illite, a = 2.3*10$^{-7}$ µM$^{-1}$ s$^{-1}$ and b = 4.8*10$^{-7}$ s$^{-1}$ and for hematite, a = 9.5*10$^{-9}$ µM$^{-1}$ s$^{-1}$ and b = 3.0*10$^{-8}$ s$^{-1}$.

The reference case is simulated under present day climate conditions and includes the enhancement of oxalate on the rate of soluble Fe formation in cloudy gridboxes only while Equation 1 is applied at every atmospheric model gridbox. Separate tracers are included for dust and for total and soluble iron in dust for Fe$_{med}$ and Fe$_{slow}$ as well as tracers for total and

soluble combustion iron. Calcite is advected to estimate the buffering effect of dust on the pH dependence of the dissolution scheme at each atmospheric model gridbox (8 new species x 4 bins + 9 = 41 total tracers added here compared to the original model which has 5 species carried as 13 total tracers to simulate aerosols for climate interactions).



### 2.5 Sensitivity Studies

To investigate the impact on soluble Fe formation of various parameters, we include six sensitivity studies (Table 1a). In the first experiment (SS1), the oxalate mechanism is removed and the formation of Fes depends solely on the proton-promoted
dissolution rate; the basis for this was to determine if the additional complexity of the oxalate scheme was required to match observations of labile iron. The second sensitivity study (SS2) is identical to the reference case but with model oxalate concentrations estimated from sulfate and was performed because other studies of Fe dissolution use sulfate as a proxy for estimating model oxalate concentrations, and follows Eq. (4):

$$[C_2O_4^{2-}]_{i,j,k} = 15 \times \frac{[SO_4^{2-}]_{i,j,k}}{\max[SO_4^{2-}]} \tag{4}$$

in units of μM. The third sensitivity study (SS3) investigates how important our dissolution mechanism is by comparing to a very simple dissolution scheme from (Hand et al., 2004). In this scheme, soluble iron dissolution depends upon cloud presence as parameterized by Equation 2 in (Hand et al., 2004). Total iron is assumed to be 3.5% of dust and Fes is
calculated from kappa*(Fet-Fes) at each cloudy gridbox, where kappa is defined in Eq. 2 in (Hand et al., 2004) as $\sum_{i,j} \frac{C_{i,j}}{C_{avg}} \frac{1}{\tau_{cld,sol}}$: $C_{i,j}$ is the cloud fraction at each atmosphere gridbox, $C_{avg}$ is the average cloud fraction at 10°N, and $\tau_{cld,sol}$ is the soluble decay lifetime (Siefert et al., 1998;Saydam and Senyuva, 2002;Hand et al., 2004). To account for the typically higher solubilities associated with combustion iron, the dissolution rate is increased by a factor of 5. The total number of tracers for SS3 is 20 (Fet_dust, Fet_comb, Fes_dust, Fes_comb and dust in each of the four size bins. In the fourth sensitivity study
(SS4), the spatial dependence of iron on mineralogy is removed and we apply the global average fraction from the reference case for calcite, Fet_med, Fet_slow and Fes_med at emission. The fifth and sixth simulations (SS5 and SS6) are identical to REF and SS2, respectively; however, pH is set to 1 instead of 2 when sulfate concentrations are greater than calcite concentrations. Because recent studies have identified highly acidic aerosol solutions (Weber et al., 2016;Guo et al., 2016), we wanted to test if the increase in the iron dissolution would better match observations of labile iron solubility.

### 2.6 Preindustrial simulations

Four preindustrial simulations are performed using the iron dissolution mechanism described in section 2.4. The first simulation (PI1) has preindustrial chemical emissions (1850) but current climate dust and combustion sources. The second simulation (PI2) includes preindustrial chemistry and dust sources with current combustion. PI3 follows PI2 with
preindustrial combustion sources and PI4 is identical to PI3 with the sulfate proxy used for calculating model oxalate concentrations instead of the OC+SOA proxy (Table 1).



### 2.7 Comparison to observations

Observations of total and soluble iron are complicated due to various collection methods (e.g. using different collection substrates and sampling with or without particle size segregation), different filter sizes used to define the soluble iron fraction, different solvents used for extraction (which determines the species of iron defined as soluble), along with myriad

definitions of the form of iron which is in fact soluble. The majority of observations are collected during cruise campaigns (daily means) as surface concentrations or dry deposition of particulate matter. These sampling campaigns inherently fail to capture the ephemeral nature of aerosol transport over ocean basins, varying in both space and time. Aerosol collections are later leached in one or more solvents to extract the soluble iron fraction from the insoluble component. A handful of alternative collection methods have been used, including the sampling of aerosols at 70m above ground level (Winton et al.,

2015) over periods of one to seven weeks, the ACE-Asia campaign that sampled total suspended particles at the surface (Chuang et al., 2005) during dust storms, and the collection of rainwater (wet deposition) at 2m above ground in the remote southern ocean (Heimburger et al., 2013). Solvents used for extraction range from acidic solutions, ultra pure water to alkaline seawater, with obvious differences in soluble iron yield due to the differences in acidity. The definition of soluble iron is rather ambiguous in part due to the field of study (i.e. oceanography, atmospheric science), the leaching solution and

the oxidation state of dissolved iron. In this study, we attempt to compare our model only to observations of labile iron, which we define as dissolved iron in either the ferrous or ferric oxidation state.

      Fine mode and coarse mode total and labile iron in the model is compared to available published observations from (Baker et al., 2006b;Baker et al., 2006a;Chen and Siefert, 2004) along with previously unpublished fine and coarse mode total and labile iron observations from Clifton Buck (personal communication, 2013, 2017). Bulk total and labile iron

observations that do not distinguish particle sizes are from Supplementary Table 2 provided in (Mahowald et al., 2009), from (Sholkovitz et al., 2012), and from (Baker et al., 2016). A set of metrics is developed to compare model simulations with the available observations to determine which set of parameterizations is necessary. The root mean square error (RMSE) is calculated for model versus observations as $\sqrt{\frac{\sum_{i=1}^{n}(mod_i - obs_i)^2}{n}}$ for total iron, labile iron and solubility (fine, coarse and fine+coarse). Similarly, the Pearson correlation coefficient is calculated between the model and observations. As a main

metric in evaluating model performance, we develop a semi-quantitative metric defined as the mean of model values at gridboxes with observations divided by the mean of the observations and will hereafter refer to this metric with the acronym MMO. The MMO does not intend to evaluate the model's ability to capture observational variability but simply to assess if the model is within the range of the observational mean.

## 3 Results

### 3.1 Comparison of observations to the reference case



In order to assess the ability of the model to simulate total, soluble iron and fractional iron solubility, model results are compared to available observations for fine (diameter < 1 μm), coarse (1 μm ≥ diameter ≤ 10 μm) and bulk (all sizes) mode particles as most available observations do not differentiate particle size. The total iron at emission is calculated based on the distribution of minerals containing iron (Journet et al., 2008;Ito and Xu, 2014;Scanza et al., 2015); Fet is approximately

3.2% of dust which is comparable to the assumption that 3.5% of dust is comprised of iron (Hand et al., 2004). Simulated surface concentrations of total iron in the fine and coarse mode are under and over-predicted, respectively compared to observations. Because 98% of Fet has prescribed diameters > 1μm, Fet is overestimated in the tropical and northern latitudes (near source regions) and underestimated below 15°S (Fe < 1μm is more likely to undergo long-range transport). Because there are relatively few observations that distinguish particle size distribution of iron, the significance of comparing

total iron surface concentrations is difficult to assess; however the model is underestimating the dynamic range in the observations (Figure S1). This is likely due in part to the fact that we are comparing annually averaged concentrations to observations that are influenced by daily weather fluctuations. Because the dust model used in this study was optimized to best capture observations of particle size distribution, deposition and aerosol optical depth (Figures 8,9 in Albani et al., 2014), surface concentrations of dust are generally over-predicted (Albani et al., 2014), which is a typical feature in dust

models (Huneeus et al., 2010). Hence it is not possible to simultaneously match dust deposition, AOD and surface concentrations. Separating total and soluble iron allows us to identify these signals and to evaluate the fractional iron solubility.

Fine mode labile iron (Fes) surface concentrations are within the range of the observations in the tropical and northern latitudes and are underestimated off the coast of Patagonia; a significant limitation arises from the lack of

observations in the regions where iron deposition should have the largest impact (see Section 3.3). Coarse mode Fes is considerably over-predicted by the model, which is consistent with the model bias in dust surface concentrations, resulting in an overestimate in the total concentration of Fes (Figure S2) (Individual fine and coarse mode Fes distributions not shown). Therefore, the fractional iron solubility at the model surface provides a more meaningful comparison to observations and facilitates the evaluation of the iron processing mechanism. In addition, examining total iron and percent solubility

separately allows us to evaluate the distribution and the solubilization of iron as distinct processes, thereby simplifying the identification and resolution of model biases in these two processes. However, additional uncertainty is introduced by considering the distributions of Fet and %Fes separately since, inherently, they are coupled. For example, erroneously high dust emissions would result in fractional solubilities that are biased low because of this coupling.

When considering percent labile iron, theory and observations support lower solubilites near dust sources where pH

is likely higher (due to the presence of calcite in dust) and where lithogenic iron has not yet undergone significant atmospheric processing; in contrast, because combustion sources are typically associated with emissions of acidic species, we expect the solubility of anthropogenic aerosols to be higher (Li et al., 2017). However, because dust iron emissions are an order of magnitude greater than emissions of combustion iron, lower solubilites near coastal regions should be dominant. In particular, we hypothesize that the simulations will predict higher percentages of Fes over remote ocean regions and





indeed, our simulations are consistent with our hypothesis; this is in line with theory and observations that show the gravitational settling of larger, less soluble iron containing aerosols near their source and long-range transport of smaller aerosols enabling multiple cycles of evaporation and condensation (Jickells et al., 2005;Baker and Croot, 2010). Simulated %Fes < 1μm is small off the coast of North Africa and Patagonia and is an order of magnitude higher in the equatorial

Pacific and southern latitude ocean regions (Figure 3b). The relatively few observations indicate reasonable agreement with the model near and directly downwind of North Africa; however there are few observations to substantiate the predictions of higher solubilities in the HNLC regions (Figure 3a). One standard deviation from the mean of daily averaged model concentrations indicates that the model is within range of the observations (Figure 3c). The spatial distribution of coarse mode %Fes is similar to the fine mode, and again our comparison is limited by the lack of observations (Figure 4).

Simulations of the sum of the fine and coarse mode %Fes show greater spatial heterogeneity in the Southern Hemisphere and less variation in Northern Hemisphere (Figure 5). While the scatterplot comparing annually-averaged %Fes versus daily cruise-based observations suggests that the model is not capturing the range of the observations, the daily averaged standard deviations for each model value indicates that our mechanism can generally capture the observations. Thus, with MMO equal to 0.8, we conclude that the soluble iron processing mechanism employed is a reasonable representation of the

atmospheric processing that dust and combustion aerosols undergo during transport.

### 3.2 Evaluation of sensitivity studies

Atmospheric iron dissolution is very complex and required many simplifications and assumptions in order to simulate the processes involved. In this study, we develop a set of metrics to evaluate the strengths and weaknesses of our proposed iron

dissolution scheme and to assess its ability to capture the true processing of atmospheric aerosols. For each simulated variable, we compare the results to available observations and calculate the RMSE (Table 2), the MMO (Table 3) and the correlation coefficient (Table 4) for the reference case and for each sensitivity study. When considering RMSE for fractional iron solubility for each case, REF compared to SS2 (oxalate calculated with a sulfate proxy), SS4 (no dependence on mineral spatial distribution at emission) and SS5-6 (higher acidity) have similar errors. SS1, the case without oxalate processing and

SS3, which corresponds to the very simple dissolution method from (Hand et al., 2004) are both associated with larger errors, indicating that the very simple mechanism (SS3) and mechanism solely based on acidic processing (SS1) are not sufficient representations. The correlation coefficients across all cases are weak, especially for the fractional iron solubilities. While the oxalate mechanism derived from the sulfate proxy (SS2) has marginally stronger correlations compared to the reference case, the difference between these is not statistically significant.

30        In order to counter the ambiguity surrounding which mechanism best matches observations, we believe the semi-quantitative MMO metric is more meaningful. A ratio greater than 1 indicates that the model is over-predicting iron while a ratio less than 1 signifies an underestimation compared to the mean of the observations. The reference case was chosen from many different simulations with varying parameters after carefully reviewing the spatial distributions of soluble iron as well as the model metrics. Although SS5 and SS6 most closely match the observational mean for total fractional iron solubility,





the pH when the concentration of sulfate is greater that calcite (pH=1) is not typically considered realistic in atmospheric waters. Recent studies that identify highly acidic aerosol solutions warrant more investigation as to whether SS5 and SS6 are in fact better representations of atmospheric iron processing. SS1 and SS3 greatly underestimate the observed mean for fractional iron solubility leading to the conclusion that the oxalate enhancement within an iron dissolution scheme is a vital

inclusion.

### 3.3 Regional and global iron deposition to ocean basins

Soluble iron deposition to global ocean basins from dust and combustion aerosols is a significant source of new iron, a limiting nutrient in many regions. This represents an important biogeochemical process that may indirectly impact climate

via a cooling effect by promoting the uptake of atmospheric $CO_2$. The global ocean is divided into twelve regions (Table S4) defined by (Gregg et al., 2003) in order to quantify deposition to ocean regions that are most likely to be iron limited. Annual-averaged global deposition of total and soluble iron from dust to the oceans is 17.4 and 0.54 Tg yr$^{-1}$, respectively and corresponds to the dissolution of 3.1% of dust iron during atmospheric transport. Including the contribution from combustion iron increases the soluble iron deposition to 3.3% of total iron. This is within the ranges reported (model: 4 ±

2%, obs.: 6 ± 8%) for global soluble iron deposition from (Ito and Xu, 2014). The soluble iron deposition for SS1 is much smaller (1.2%) indicating that including the oxalate scheme increases the amount of Fes deposition by nearly three times.

Iron is a limiting nutrient in remote ocean regions, with these basins defined as follows: the Antarctic basin (ocean regions south of 30°S), the southern Atlantic, Pacific and Indian Ocean basins (30°S - 10°S), and the equatorial Pacific basin (Table S4). These five basins comprise around 60% of the global ocean but only receive 3.5% of dust deposition, which is

the main source of new iron to these regions (Table 5a). The Antarctic basin receives just 2.4% of the globally deposited bioavailable iron, the south Indian Ocean, Pacific and Atlantic basins receive 0.9, 0.4 and 1.3% of bioavailable iron and the equatorial Pacific basin receives 1.5%; as such, the global HNLC regions receive just a small fraction of the total bioavailable iron deposited to the ocean surface (6.5%). The total fractional iron solubility (dust and combustion sources) for the equatorial Pacific, the south Atlantic, Pacific and Indian Ocean, and the Antarctic basins is largely influenced by the

contribution from combustion (Table 5b). In general, we observe that the percent of bioavailable iron reaching different ocean basins is significantly affected by both the location of clouds (e.g. oxalate) and combustion sources. For example, SS2, the simulation where global oxalate concentrations are calculated via the sulfate proxy rather than the organic carbon proxy results in decreased fractional iron solubility in the southern hemisphere and equatorial ocean basins and increased solubility in the northern basins. The latitudinal shift in the spatial distribution of modeled oxalate concentrations (Figure 2)

clearly explains the spatial differences in the labile iron percentages reaching the different ocean basins.

The spatial and zonal distributions of labile iron production lifetimes, defined as the difference between Fet and Fes divided by the total Fes production, (Figure 6) illustrate where and how dust and combustion iron is most efficiently processed. Shorter lifetimes reflect where most of the atmospheric processing occurs which is in geographic areas low in calcite, high in oxalate and combustion aerosols, and in regions dominated by clouds and precipitation (e.g. equatorial




regions). Iron dissolution is much slower near dust sources where acidic processing is hindered by the pH buffering of calcite in dust; additionally dust source regions are characterized by arid surface conditions and low cloud coverage making the in-cloud oxalate processing insignificant. In spite of the production lifetimes, because the total iron from dust is 30 times larger than total combustion iron, the spatial distribution of labile iron deposition is dominated by dust iron (Figure 7a).

Again, however, the fractional iron solubility is inversely related to total and soluble iron deposition, a result consistent with theory and observations (e. g. (Sholkovitz et al., 2012)).

### 3.4 Comparison to sensitivity studies

Six sensitivity studies are conducted to assess the importance of the assumptions made for the reference case. The

atmospheric burden, wet and dry deposition and emission of dust, total iron in dust, $Fes_{med}$, $Fes_{slow}$, $Fet_{comb}$, and $Fes_{comb}$ are compared between the sensitivity simulations and the reference case (Table S2). Dust and $Fet_{comb}$ loading and emission are identical for all cases. The combined wet and dry deposition of soluble iron is about 2.5 times higher for the reference case compared to SS1 and over 6 times greater compared to SS3. This suggests that a more complex mechanism than the simple scheme in SS3 is needed; additionally, the SS1 simulation strongly suggests the need for including the organic ligand

enhancement in our iron dissolution scheme. SS5 and SS6 are identical to REF and SS2 with the exception of higher acidity prescribed when the concentration of sulfate exceeds calcite. The increased acidity corresponds to greater iron dissolution; while RMSE of fractional iron solubility (Table 2) is similar for these studies, the MMO is considerably improved for fine mode and total solubility (Table 3). Despite more closely matching the observational mean, a pH of 1 is typically considered unrealistic; however, more recent studies have identified conditions yielding very acidic atmospheric water, particularly in

fine particle solutions (Guo et al., 2016;Weber et al., 2016). The largest differences in soluble iron deposition and %Fes between the reference case and the six sensitivity studies are for decreasing the complexity of the iron processing mechanism (SS1, SS3) and for increasing the acidity (SS5, SS6).

Because oxalate formation in the atmosphere is complex, some iron mobilization studies that calculate oxalate concentrations use a sulfate proxy for several reasons (Johnson and Meskhidze, 2013). The atmospheric sulfate

concentrations and sources of sulfate are better understood than OC and especially SOA concentrations, and typically occur near combustion sources; a formula for the concentration of oxalate as a function of sulfate developed in (Yu et al., 2005) is used for studies that want to include the impact of organic ligand enhancement (Johnson and Meskhidze, 2013). SS2 uses a similar proxy and we conclude that the method for parameterizing model concentrations of oxalate is not important for matching observational means in our model compared to other factors (Table 7). The spatial distribution for this analysis

supports the above discussion (Figure 8) and highlights the regional differences in Fes deposition particularly between the two different parameterizations for oxalate (Figure 8b), which can account for up to a 30% difference in soluble iron fluxes depending on which proxy is used (Table 6b). Despite these differences, it is reasonable to use either proxy for calculating oxalate; we chose to use the OC proxy since the spatial oxalate distributions generated from this more closely match distributions of explicitly simulated oxalate in (Myriokefalitakis et al., 2011).





In order to assess the importance of deriving global iron emission maps from iron-containing mineral soil maps, the mineral spatial dependence on iron emission is removed for SS4 and each tracer is prescribed a global average fraction for total iron, soluble iron and calcite. These fractions are determined from the global average surface fluxes of each tracer from the reference case. The MMO for this case shows that SS4 underestimates fine mode fractional iron solubility, matches this

in the coarse mode, and underestimates the combined fine and coarse mode solubility with the observational mean. Additionally the global average fractional iron solubility is marginally smaller (11%, Table 7). Again, the differences in the global budgets between REF and SS4 are not large in absolute values, although for different basins it could change by up to 20% (Table 6b).

**4 Results: Preindustrial studies**

Four preindustrial simulations are conducted using the iron processing mechanism from the reference case. PI1 has the same dust and combustion sources as the current climate runs but is forced with preindustrial (1850) chemistry. PI2 includes the preindustrial chemistry as well as the preindustrial dust sources; PI3 includes preindustrial combustion emissions. The fourth simulation has the changes included in PI3 and is run with oxalate parameterized via the sulfate proxy from SS2.

Previous studies based on limited paleodata suggest that dust emissions in 1850 were significantly lower than for current climate (Mahowald et al., 2010;Mulitza et al., 2010). In addition, industrial combustion sources in 1850 are considerably lower that present-day emissions (Lamarque et al., 2010). Annually averaged global deposition of labile iron from both dust and combustion is 2X higher for current climate (REF) compared with PI3 (Table S2,3), while labile iron deposition to all ocean basins and to HLNC basins is ~1.7 and 1.5 (Table 5,6, ocean basin deposition for PI not shown) times

higher than preindustrial. The ratio of REF/PI3 for soluble iron deposition is shown in Figure 9b where over most of the globe, more soluble iron is deposited for current climate. Interestingly, in some regions, combustion iron from biomass burning is significant in PI3, particularly in South America. This increase actually results in higher percent labile iron deposition to the Antarctic Ocean basin during the preindustrial compared to current climate (Table 6b); the average contribution of combustion iron to total fractional iron solubility in the ocean basins below 10°S is 45% for current climate

and 55% for preindustrial conditions.

The ratio of PI3/PI4 has a similar spatial distribution to Figure 8b although lower overall acidity and combustion in 1850 decreases the magnitude of the differences. However, if one considers the change in soluble iron between current and preindustrial, a strong hemispheric gradient appears in the ratio between using $SO_4^{2-}$ or OC as the proxy for oxalate because of the changes in these constituent precursors in the input data (Figure 9c). The differences are a 20% decrease and a 20 %

increase in the northern and southern hemispheres, respectively. Thus the choice of SO4 or OC as the proxy for oxalate might not cause a statistically significant change when compared to observations, but it has implications for projected changes due to changes in precursor emissions.

**5 Discussion and comparison to previous studies**





This study compares observations of soluble iron with an iron dissolution scheme that is simple enough to include in an Earth system model and includes chemical mechanisms thought to be important in atmospheric iron processing based on studies that incorporate coupled, complex chemical schemes (Meskhidze et al., 2005;Solmon et al., 2009;Johnson and Meskhidze, 2013;Ito and Xu, 2014;Myriokefalitakis et al., 2015;Ito, 2015). Currently, there is substantial uncertainty

regarding the relative importance of how the various atmospheric processes responsible for iron dissolution interact with each other, the significance of and processing of combustion iron, and the bioavailability of atmospherically processed iron following deposition to the ocean.

A comparison of our results with the results obtained from previous modeling studies enables some determination of the robustness of the iron dissolution scheme used in this study. SS1 is compared to a similar study (Ito and Xu, 2014)

since this study did not include an oxalate scheme. The deposition of total and soluble iron from dust and combustion sources is calculated for ocean regions in the western and eastern North Pacific in Table S2e and compared to Table 6 in (Ito and Xu, 2014). Because we used similar definitions for Fet and a similar dissolution scheme for acidic processing of iron, our total and soluble iron reasonably matches their data. In both the western and eastern North Pacific basins, Ito and Xu (2014) report dust iron solubility of around 2%; we calculate dust iron solubility of around 1.4 and 1.6% for the two basins,

respectively. This disparity may be a result of not including separate tracers and dissolution rates for $Fe_{RR}$ along with some simplifications we made in the dissolution rate (i.e. not accounting for the deviation from equilibrium for Fet→Fes). The sources of combustion in this study are different and we did not include a combustion-specific processing scheme. For lack of a better understanding of combustion iron dissolution, combustion iron in this study is mobilized with the rate used for $Fes_{med}$.

Johnson and Meskhidze (2013) include an oxalate processing scheme based on oxalate promoted dissolution data reported in (Paris et al., 2011). It is difficult to compare our results since we interpreted the Paris et al., 2011 data differently to calculate our coefficients for the oxalate dissolution rates for illite and hematite. We predict higher dissolved iron deposition from dust to the global ocean basins compared to their prediction (0.54 versus 0.21 Tg yr$^{-1}$) and for the range in Fes deposition reported here (0.21-0.41 Tg yr$^{-1}$) from (Luo et al., 2008;Luo and Gao, 2010;Okin et al., 2011). Johnson and

Meskhidze (2013) report a 75% increase in Fes deposition when the oxalate mechanism is included, which is similar to our prediction of a 63% increase (Table 6b).

A recent study that included both dust and combustion iron along with acidic processing, photochemical reduction and organic ligand processing (Myriokefalitakis et al., 2015) reports Fet emissions from mineral dust and combustion at ~35 Tg yr$^{-1}$ and 2 Tg yr$^{-1}$, respectively. In fact, the method by which this study determined the iron content at emission is nearly

identical to the method used in our study; we calculate ~57 and 2 Tg yr$^{-1}$ of total iron, which is very similar to their emissions if we account for the 38% difference in dust emission between the two studies. However, the global annual flux of Fet from dust and combustion in our study is nearly 3x higher than the 0.496 Tg yr$^{-1}$ reported here. A significant difference between our mechanism involves the oxalate processing; we assume that medium soluble iron (iron from illite and smectite) undergoes organic ligand processing with a rate determined from Paris et al., 2011 for illite and slow soluble iron



(iron from kaolinite, feldspar, and hematite) is processed with a rate derived from the hematite data. This study only processes illite and hematite and neglects the iron from smectite, feldspar and kaolinite, the first of which has significant iron concentrations as reported in (Journet et al., 2008). In addition, because we used a simple parameterization for pH, we do not simulate the decrease in the oxalate mechanism at low pH. Regardless, our ability to match observations, despite the

simplified parameterizations we employed compared with the full chemistry scheme used here, is at least as good if not better than the metric values reported in the supplementary material.

Ito (2015) includes an oxalate processing mechanism for the dissolution of iron in combustion aerosols. The global annual deposition for labile iron reported is 1.07 Tg yr$^{-1}$, which is closer to the 1.4 Tg yr$^{-1}$ estimated in this study.

Theory and observations of iron processing show an inverse trend in total and soluble iron concentrations with

fractional iron solubility (Sholkovitz et al., 2012). The results from our work suggest that this relationship cannot be uniquely constrained by considering either dust or combustion sources of iron alone. Figure 10 shows the total fractional iron solubility versus total iron for dust iron and combustion iron. In order to illustrate the difficulty in determining which process is responsible for the inverse trend, for each observation, we define a box spanning the standard deviation of modeled fractional iron solubility and of modeled total iron. When the observation lies inside or outside of this box, we can

calculate the percentage of model points that are within the range of the observations. For dust processing only, for combustion only and for the total (dust+combustion), 75.4, 73.8 and 73.3% of model values are within the observational range, respectively. If only combustion sources were included and assumed to be more soluble, the modeled values for soluble iron would all be shifted upwards in Figure 10 and could match available observations quite well.

**6 Conclusion**

A medium-complexity soluble iron processing mechanism is developed for use in an Earth system model and includes a proton-promoted aerosol water processing scheme along with an in-cloud organic ligand-promoted dissolution scheme. We define a semi-quantitative metric as the model mean at points with observations divided by the observational mean (MMO); fractional iron solubility MMO is 0.8, indicating that while the model is not capturing all of the observational variability, it is

within range of the observational mean. Several sensitivity studies are performed to ascertain the degree of complexity needed to match observations. A simple first order decay rate from (Hand et al., 2004) does much worse compared to observations than our intermediate complexity mechanism. Our results indicate that the acidic processing alone is insufficient in matching observations and that the additional ligand-promoted mechanism was required; the mechanism employed is reasonably able to capture some but not all of the observational features of soluble iron and iron solubility,

similar to the more complicated mechanisms employed in previous studies (Meskhidze et al., 2005;Johnson and Meskhidze, 2013;Ito and Xu, 2014;Myriokefalitakis et al., 2015;Ito, 2015). The in-cloud organic ligand processing mechanism is likely to be responsible for the majority of the atmospheric processing (>60%).

An additional objective was characterizing the relative importance of the contribution of iron from combustion aerosols versus iron from dust to different ocean basins. Upon deposition to global oceans, global mean combustion iron



solubility to total fractional iron solubility is 8.2%; however, the contribution of fractional iron solubility from combustion sources to ocean basins below 15°S is approximately 50%. We estimate approximately a doubling of soluble iron deposition to the oceans in the current climate relative to the preindustrial due to increased dust, combustion iron and acidifying compounds, and a ~1.7x-fold increase in the iron-limited regions of the ocean (Tables 5-7). Even in the preindustrial, when

industrial sources of iron are much lower, combustion sources of iron are likely to be important in some regions.

Building the framework to model iron dissolution in a full complexity Earth system model was the main goal of this study; however, more observations are needed in remote ocean regions to evaluate model performance. Additionally more laboratory and field experiments are necessary to better constrain the importance of combustion iron and atmospheric processing of mineral aerosols with both inorganic and organic acids. The sensitivity studies indicate that including an

organic ligand processing mechanism is the most important parameter in order to match observations. Further investigation into atmospheric aerosol solution acidity is necessary in order to distinguish whether the sensitivity studies with higher acidity are better representations of soluble iron processing in the atmosphere.

**Acknowledgments**

We would like the acknowledge the support of DOE DE-SC0006735 and DOE DE-SC0006735 and NSF 1049033. We would also like to thank Douglas Hamilton for comments on the text. Carlos Pérez García-Pando acknowledges long-term support from the AXA Research Fund through an AXA Chair on Sand and Dust Storms, as well as the support received through the Ramón y Cajal programme (grant RYC-2015-18690) of the Spanish Ministry of Economy and Competitiveness.

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





**Tables**

**Table 1a:** List of current climate simulation used in this study. $[H^+]$ indicates proton-promoted aerosol water iron processing. $C_2O_4^{2-}{}_{OC\text{-}proxy}$ is the parameterization for in-cloud iron processing including the oxalate parameterization derived from model concentrations of organic carbon and secondary organic carbon. $C_2O_4^{2-}{}_{SO4\text{-}proxy}$ calculates oxalate concentration from model sulphate concentrations.

|  | $[H^+]$ | $C_2O_4^{2-}{}_{OC\text{-}proxy}$ | $C_2O_4^{2-}{}_{SO4\text{-}proxy}$ | pH |
|---|---|---|---|---|
| **REF** | X | X |  | 2 |
| **SS1** | X |  |  | 2 |
| **SS2** | X |  | X | 2 |
| **SS3** | n/a | n/a | n/a | n/a |
| **SS4** | X | X |  | 2 |
| **SS5** | X | X |  | 1 |
| **SS6** | X |  | X | 1 |

**Table 1b:** List of preindustrial simulations used in this study. Preindustrial simulations assume pH = 2 when model concentrations of sulphate are greater than calcite.

|  | $[H^+]$ | $C_2O_4^{2-}{}_{OC\text{-}proxy}$ | $C_2O_4^{2-}{}_{SO4\text{-}proxy}$ | Emissions | Dust-source | PI-comb |
|---|---|---|---|---|---|---|
| **PI1** | X | X |  | X |  |  |
| **PI2** | X | X |  | X | X |  |
| **PI3** | X | X |  | X | X | X |
| **PI4** | X |  | X | X | X | X |





**Table 2:** Root mean square error (RMSE) for the reference case (REF) and for sensitivity studies 1-6. REF: proton promoted and oxalate promoted iron dissolution; SS1: proton promoted iron dissolution; SS2: REF with oxalate dependent on sulphate; SS3: simple cloud parameterization for iron dissolution (Hand et al., 2004); SS4: REF with no spatial dependence on mineralogy; SS5: REF, pH = 1; SS6: SS2, pH = 1.

| RMSE | REF | SS1 | SS2 | SS3 | SS4 | SS5 | SS6 |
|---|---|---|---|---|---|---|---|
| fes fine | 5.9 | 6.1 | 5.9 | 6.6 | 6.2 | 5.9 | 5.9 |
| fes coarse | 26.5 | 13.8 | 27.9 | 5.3 | 20.7 | 27.2 | 28.6 |
| fes total | 29.0 | 20.7 | 30.2 | 16.2 | 23.1 | 29.5 | 30.7 |
| fet fine | 190.6 | 190.6 | 190.6 | 193.2 | 194.1 | 190.6 | 190.6 |
| fet coarse | 1448.4 | 1448.6 | 1448.0 | 1356.6 | 1241.8 | 1448.4 | 1447.9 |
| fet total | 1516.0 | 1516.2 | 1515.9 | 2450.5 | 1400.0 | 1516.1 | 1515.9 |
| fes %fine | 19.5 | 20.3 | 19.4 | 20.6 | 19.9 | 19.6 | 19.4 |
| fes %coarse | 7.1 | 7.6 | 6.9 | 7.5 | 7.1 | 7.4 | 7.2 |
| fes %total | 10.0 | 11.1 | 9.8 | 10.6 | 10.1 | 10.3 | 10.1 |



**Table 3:** Mean of model at gridboxes with observations divided by the mean of the observations for the reference case (REF) and for sensitivity studies 1-6.

| MEAN(mod/obs) | REF | SS1 | SS2 | SS3 | SS4 | SS5 | SS6 |
|---|---|---|---|---|---|---|---|
| fes fine | 0.70 | 0.47 | 0.69 | 0.09 | 0.24 | 0.73 | 0.72 |
| fes coarse | 6.80 | 2.89 | 6.90 | 1.58 | 5.37 | 7.09 | 7.19 |
| fes total | 2.19 | 1.02 | 2.23 | 0.56 | 1.62 | 2.30 | 2.34 |
| fet fine | 0.33 | 0.33 | 0.33 | 0.23 | 0.21 | 0.33 | 0.33 |
| fet coarse | 4.09 | 4.09 | 4.09 | 3.72 | 3.42 | 4.09 | 4.08 |
| fet total | 1.33 | 1.33 | 1.33 | 1.22 | 1.12 | 1.33 | 1.33 |
| fes %fine | 0.85 | 0.44 | 0.82 | 0.33 | 0.64 | 0.98 | 0.95 |
| fes %coarse | 1.14 | 0.36 | 1.09 | 0.38 | 1.06 | 1.32 | 1.28 |
| fes %total | 0.80 | 0.29 | 0.79 | 0.37 | 0.74 | 0.94 | 0.93 |
| tau | 1579 | 5440 | 1569 | | 1489 | 841 | 834 |
| mean %fesdep | 3.26 | 1.20 | 3.14 | 0.86 | 2.91 | 3.49 | 3.37 |



**Table 4:** Correlation coefficients for the reference case and sensitivity studies 1-6.

| Corr. Coeff. | REF | SS1 | SS2 | SS3 | SS4 | SS5 | SS6 |
|---|---|---|---|---|---|---|---|
| fes fine | 0.324 | 0.275 | 0.319 | 0.274 | 0.349 | 0.323 | 0.318 |
| fes coarse | 0.511 | 0.442 | 0.518 | 0.520 | 0.509 | 0.511 | 0.518 |
| fes total | 0.119 | 0.091 | 0.118 | 0.267 | 0.118 | 0.119 | 0.118 |
| fet fine | 0.270 | 0.270 | 0.270 | 0.271 | 0.271 | 0.270 | 0.270 |
| fet coarse | 0.298 | 0.298 | 0.298 | 0.286 | 0.286 | 0.298 | 0.298 |
| fet total | 0.247 | 0.247 | 0.247 | 0.122 | 0.244 | 0.247 | 0.247 |
| fes %fine | 0.062 | 0.046 | 0.117 | 0.070 | 0.050 | 0.054 | 0.092 |
| fes %coarse | 0.157 | 0.155 | 0.242 | 0.233 | 0.143 | 0.153 | 0.212 |
| fes %total | 0.024 | -0.068 | 0.081 | 0.109 | 0.009 | -0.002 | 0.036 |



**Table 5a:** Global ocean and regional ocean basin total deposition for dust (Tg yr$^{-1}$), total iron from dust and combustion, labile iron in dust and combustion aerosols (Gg yr$^{-1}$).

| | Dust | Fet$_{med}$ | Fet$_{slow}$ | Fes$_{med}$ | Fes$_{slow}$ | Fe$_{comb}$ | Fes$_{comb}$ |
|---|---|---|---|---|---|---|---|
| | (Tg/yr) | | | (Gg/yr) | | | |
| **Global** | 513.3 | 13073.7 | 4278.1 | 530.1 | 7.8 | 586.1 | 47.7 |
| **N.Atlantic** | 147.5 | 3702.7 | 1074.1 | 94.8 | 1.0 | 69.5 | 4.1 |
| **N.Pacific** | 20.5 | 611.4 | 165.1 | 35.6 | 0.6 | 123.8 | 8.2 |
| **N.Cen.Atl.** | 108.3 | 2792.7 | 832.9 | 105.1 | 1.3 | 13.8 | 1.5 |
| **N.Cen.Pac.** | 6.4 | 196.7 | 53.0 | 15.5 | 0.3 | 30.3 | 2.3 |
| **N.Indian.Ocn** | 139.4 | 3180.7 | 1319.9 | 84.9 | 1.3 | 59.3 | 4.0 |
| **Equat.Atl.** | 63.0 | 1918.8 | 593.0 | 169.7 | 2.7 | 88.8 | 11.3 |
| **Equat.Pac.** | 1.8 | 60.2 | 16.2 | 6.7 | 0.1 | 15.1 | 1.7 |
| **Equat.In.Ocn.** | 30.5 | 787.5 | 216.1 | 34.5 | 0.7 | 89.0 | 6.2 |
| **S.Atlantic** | 3.0 | 69.5 | 28.7 | 4.3 | 0.1 | 38.3 | 3.3 |
| **S.Pacific** | 0.5 | 15.2 | 8.1 | 1.0 | 0.1 | 16.7 | 1.1 |
| **S.Indian.Ocn** | 1.0 | 29.1 | 12.2 | 3.0 | 0.1 | 35.6 | 2.5 |
| **Antarctic** | 11.8 | 297.6 | 128.6 | 8.5 | 0.2 | 55.3 | 5.4 |

**Table 5b:** Global ocean and regional ocean basin fractional iron solubility for fine, coarse and total dust and combustion iron, total fractional iron solubility from both dust and combustion, and the fraction of percent soluble combustion iron to total fractional iron solubility.

| | %Fe$_{fine}$ | %Fe$_{coarse}$ | %Fe | %Fec$_{fine}$ | %Fec$_{coarse}$ | %Fec | %Fes$_{d+c}$ | %Fec/Fes$_{d+c}$ |
|---|---|---|---|---|---|---|---|---|
| **Global** | 8.56 | 0.28 | 3.10 | 10.36 | 7.68 | 8.14 | 3.26 | 8.15 |
| **N.Atlantic** | 6.59 | 1.91 | 2.01 | 7.90 | 5.61 | 5.89 | 2.06 | 4.10 |
| **N.Pacific** | 8.53 | 4.47 | 4.66 | 8.01 | 6.28 | 6.60 | 4.92 | 18.44 |
| **N.Cen.Atl.** | 7.99 | 2.80 | 2.93 | 14.95 | 9.74 | 10.88 | 2.96 | 1.39 |
| **N.Cen.Pac.** | 10.93 | 0.75 | 6.35 | 9.67 | 7.15 | 7.66 | 6.49 | 12.78 |
| **N.Indian.Ocn** | 6.57 | 1.84 | 1.91 | 8.76 | 6.37 | 6.75 | 1.98 | 4.44 |
| **Eq.Atlantic** | 10.72 | 6.64 | 6.86 | 14.87 | 12.22 | 12.75 | 7.06 | 6.17 |
| **Eq.Pacific** | 13.08 | 8.62 | 8.97 | 13.71 | 10.58 | 11.33 | 9.37 | 19.99 |
| **Eq.Indian.Ocn** | 10.08 | 3.34 | 3.51 | 8.87 | 6.60 | 6.94 | 3.79 | 14.93 |
| **S.Atlantic** | 12.79 | 4.17 | 4.42 | 10.86 | 8.33 | 8.70 | 5.62 | 43.42 |
| **S.Pacific** | 10.77 | 4.16 | 4.40 | 9.72 | 6.36 | 6.78 | 5.40 | 52.44 |
| **S.Indian.Ocn** | 12.01 | 7.19 | 7.47 | 8.99 | 6.63 | 6.94 | 7.23 | 44.50 |
| **Antarctic** | 6.78 | 1.92 | 2.03 | 11.20 | 9.37 | 9.78 | 2.92 | 38.45 |





**Table 6a:** Annually averaged loading (Gg), deposition (Gg yr$^{-1}$) and fractional iron solubility at deposition to global and regional oceans for the reference case.

| | FET$_{load}$ | FES$_{load}$ | FET$_{dep}$ | FES$_{dep}$ | %FES$_{dep}$ |
|---|---|---|---|---|---|
| **Global ocean** | 201.22 | 7.21 | 17937.93 | 585.57 | 3.26 |
| **N.Atlantic** | 38.63 | 0.99 | 4846.37 | 99.87 | 2.06 |
| **N.Pacific** | 12.71 | 0.56 | 900.25 | 44.32 | 4.92 |
| **N.Cen.Atl.** | 49.83 | 1.48 | 3639.33 | 107.86 | 2.96 |
| **N.Cen.Pac.** | 5.10 | 0.29 | 280.01 | 18.17 | 6.49 |
| **N.Indian.Ocn** | 57.11 | 1.21 | 4559.84 | 90.13 | 1.98 |
| **Equat.Atl.** | 32.79 | 2.18 | 2600.55 | 183.69 | 7.06 |
| **Equat.Pac.** | 0.83 | 0.08 | 91.48 | 8.56 | 9.37 |
| **Equat.In.Ocn.** | 8.03 | 0.39 | 1092.71 | 41.41 | 3.79 |
| **S.Atlantic** | 3.21 | 0.25 | 136.42 | 7.67 | 5.62 |
| **S.Pacific** | 0.60 | 0.04 | 40.06 | 2.16 | 5.40 |
| **S.Indian.Ocn** | 1.21 | 0.08 | 76.83 | 5.55 | 7.23 |
| **Antarctic** | 2.26 | 0.12 | 481.45 | 14.08 | 2.92 |

5 **Table 6b:** Percent change for all cases relative to the reference case for global and regional fractional iron solubility at deposition to the ocean.

| | SS1 | SS2 | SS3 | SS4 | SS5 | SS6 | PI1 | PI2 | PI3 | PI4 |
|---|---|---|---|---|---|---|---|---|---|---|
| **Global** | -63.2% | -3.7% | -73.7% | -13.6% | 7.0% | 3.3% | -14.3% | -12.0% | -15.2% | -16.7% |
| **N.Atlantic** | -49.8% | 11.5% | -79.0% | -12.6% | 4.4% | 15.8% | -6.5% | -4.6% | -6.9% | -2.4% |
| **N.Pacific** | -62.8% | 9.4% | -49.6% | -9.5% | 11.5% | 20.8% | -13.4% | -15.9% | -19.7% | -20.4% |
| **N.Cen.Atl.** | -63.8% | 3.5% | -75.7% | -16.2% | 4.0% | 7.5% | -7.2% | -5.1% | -5.7% | -3.5% |
| **N.Cen.Pac.** | -63.8% | 6.8% | -59.8% | -12.2% | 20.6% | 27.3% | -13.5% | -20.4% | -22.7% | -20.6% |
| **N.Indian.Ocn** | -49.9% | 9.8% | -77.9% | -5.4% | 7.5% | 17.2% | -9.8% | -8.4% | -11.0% | -10.4% |
| **Equat.Atl.** | -80.8% | -28.5% | -76.6% | -7.6% | 4.0% | -24.4% | -26.1% | -23.6% | -25.5% | -35.0% |
| **Equat.Pac.** | -68.7% | -7.9% | -54.2% | -8.6% | 18.3% | 10.7% | -14.1% | -11.9% | -15.1% | -16.1% |
| **Equat.In.Ocn.** | -58.2% | 7.2% | -68.4% | -14.6% | 13.5% | 20.7% | -12.9% | -11.0% | -18.3% | -18.2% |
| **S.Atlantic** | -56.1% | -19.4% | -49.1% | -10.9% | 16.8% | -2.2% | -20.0% | -24.1% | -29.1% | -32.6% |
| **S.Pacific** | -42.4% | -5.4% | -24.6% | 0.8% | 23.6% | 18.5% | -9.6% | -3.5% | -6.1% | -7.1% |
| **S.Indian.Ocn** | -52.4% | -12.7% | -36.7% | -3.8% | 20.3% | 7.8% | -16.3% | -16.1% | -15.3% | -18.6% |
| **Antarctic** | -46.4% | -6.0% | -42.5% | -0.6% | 15.9% | 10.1% | -9.1% | 3.1% | 3.3% | 2.7% |





**Table 7:** Percent change for the MMO metric compared to the reference case for labile, total and percent labile iron in the fine mode (< 1 μm), the coarse mode (> 1 μm and ≤ 10 μm) and for the combined fine and coarse modes (0.1 μm – 10.0 μm). The lower section of the table compares the percent difference for the average production lifetime of Fes (days) and the average fractional iron solubility at deposition to global ocean basins.

| MEAN(mod/obs) | REF | SS1 | SS2 | SS3 | SS4 | SS5 | SS6 |
|---|---|---|---|---|---|---|---|
| fes fine | 0.70 | -32.3% | -1.4% | -87.1% | -66.1% | 4.5% | 3.1% |
| fes coarse | 6.80 | -57.5% | 1.5% | -76.8% | -21.1% | 4.2% | 5.8% |
| fes total | 2.19 | -53.4% | 1.9% | -74.5% | -26.2% | 4.8% | 6.8% |
| fet fine | 0.33 | 0.0% | 0.0% | -30.8% | -36.1% | 0.0% | 0.0% |
| fet coarse | 4.09 | 0.0% | 0.0% | -9.0% | -16.4% | 0.0% | 0.0% |
| fet total | 1.33 | 0.0% | 0.0% | -8.4% | -16.1% | 0.0% | 0.0% |
| fes %fine | 0.85 | -47.9% | -4.4% | -61.2% | -25.5% | 15.1% | 10.8% |
| fes %coarse | 1.14 | -68.4% | -4.2% | -66.4% | -7.1% | 16.3% | 12.1% |
| fes %total | 0.80 | -63.4% | -1.6% | -54.0% | -7.5% | 18.4% | 16.9% |
| tau | 1579 | 244.6% | -0.6% | n/a | -5.7% | -46.7% | -47.2% |
| mean %fesdep | 3.26 | -63.2% | -3.7% | -73.7% | -11.0% | 7.0% | 3.3% |





**Figure captions**

**Figure 1:** Iron dissolution rate for illite and hematite representing medium soluble iron and slow soluble iron respectively. '+' indicates the proton-promoted dissolution rate. Solid, dashed and dotted lines are the addition of the proton-promoted and the oxalate-promoted dissolution rates at [oxalate] = 0, [oxalate] = 1 and [oxalate] = 10 μM, respectively.

**Figure 2:** Spatial distribution of annual averaged surface oxalate concentrations (μM). Oxalate concentrations are derived from modeled organic carbon concentrations **(a)** and from modeled sulfate concentrations **(b)**.

**Figure 3:** Observations of **(a)** and modeled surface concentrations **(b)** of percent labile iron for particle diameter < 1μm. Scatterplot of observed fine mode percent labile Fe versus modeled fine mode percent labile Fe **(c)**. Red crosses indicate observations about 15°N, blue boxes indicate observations between 15°S and 15°N and green diamonds indicate observations below 15°S. Grey vertical lines correspond to one standard deviation of model daily averaged concentrations between 2007 and 2011.

**Figure 4:** Observations of **(a)** and modeled surface concentrations **(b)** of percent labile iron for particle diameter > 1μm Scatterplot of observed coarse mode percent labile Fe versus modeled coarse mode percent labile Fe **(c)**. Red crosses indicate observations about 15°N, blue boxes indicate observations between 15°S and 15°N and green diamonds indicate observations below 15°S. Grey vertical lines correspond to one standard deviation of model daily averaged concentrations between 2007 and 2011.

**Figure 5:** Observations of **(a)** and modeled surface concentrations **(b)** of percent labile iron. Scatterplot of observed fine mode percent labile Fe versus modeled fine mode percent labile Fe **(c)**. Red crosses indicate observations about 15°N, blue boxes indicate observations between 15°S and 15°N and green diamonds indicate observations below 15°S. Grey vertical lines correspond to one standard deviation of model daily averaged concentrations between 2007 and 2011.

**Figure 6**: Spatial distribution of annually averaged turnover time of insoluble iron (days) defined as the total insoluble iron from dust and combustion aerosols divided by the production of soluble iron from insoluble iron **(a)**. Zonal distribution of annually averaged insoluble to soluble iron turnover (days) **(b)**.

**Figure 7:** Spatial distribution of annually averaged soluble iron deposition from both dust and combustion in Tg yr$^{-1}$ **(a)**. Spatial distribution of annually averaged fractional iron solubility from dust and combustion (%) **(b)**.

**Figure 8**: The spatial distribution of the Fes deposition ratio of the reference case over the six sensitivity studies. SS1 corresponds to the case with proton-promoted iron dissolution and no processing via organic ligand dissolution enhancement. The processing mechanism for SS2 is identical to the reference case however uses a sulfate proxy for calculating model oxalate concentrations instead of the OC proxy used in the reference case. SS3 corresponds to a simplified dissolution mechanism defined in Hand et al., 2004; a decay rate is applied when iron comes into contact with a cloud. The SS4 simulation has the same processing mechanism as the reference case however the spatial dependence of total and soluble iron on mineralogy at emission is replaced with the average values for these variables from the reference case. SS5 and SS6 are identical to the reference case and SS2, respectively, with the exception of prescribing the pH at 1 when sulfate concentrations exceed calcite concentrations. White indicates no change, red indicates that the deposition in the reference case is higher than the sensitivity study and blue indicates higher deposition compared to the reference case.

**Figure 9:** Spatial distribution of soluble iron deposition for the preindustrial simulation with the year 1850 chemical emissions, dust source strengths and combustion emissions (PI3) (Tg yr$^{-1}$) in panel **(a)**. Spatial distribution of the Fes deposition ratio of the reference case over PI3 **(b)**, and the difference between the sulfate and OC proxies for current over preindustrial **(c)**.



**Figure 10:** Scatterplots of total iron vs. fractional iron solubility. For panels a-c, black points correspond to observations, blue circles correspond to annually averaged model values at gridboxes where we have observational data. Red stars indicate model values that fall outside the range (predicted from the average of 1 standard deviation from daily averaged values for both total iron and fractional iron solubility) in relation to observed values. Panel a corresponds to fractional iron solubility

5  from both dust and combustion, Panel b is for fractional iron solubility from dust only and panel c is for combustion iron only. Panel d corresponds to the zoomed in version of Figure 1 in Sholkovitz et al., 2012 shown logarithmically.


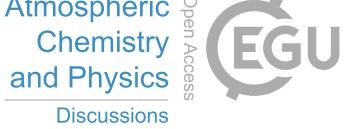

**Figure 1**

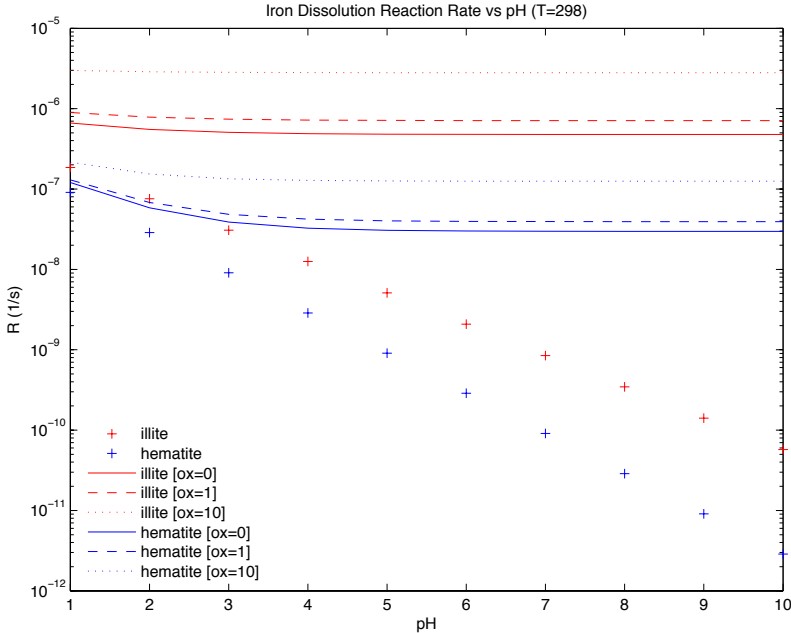





**Figure 2**

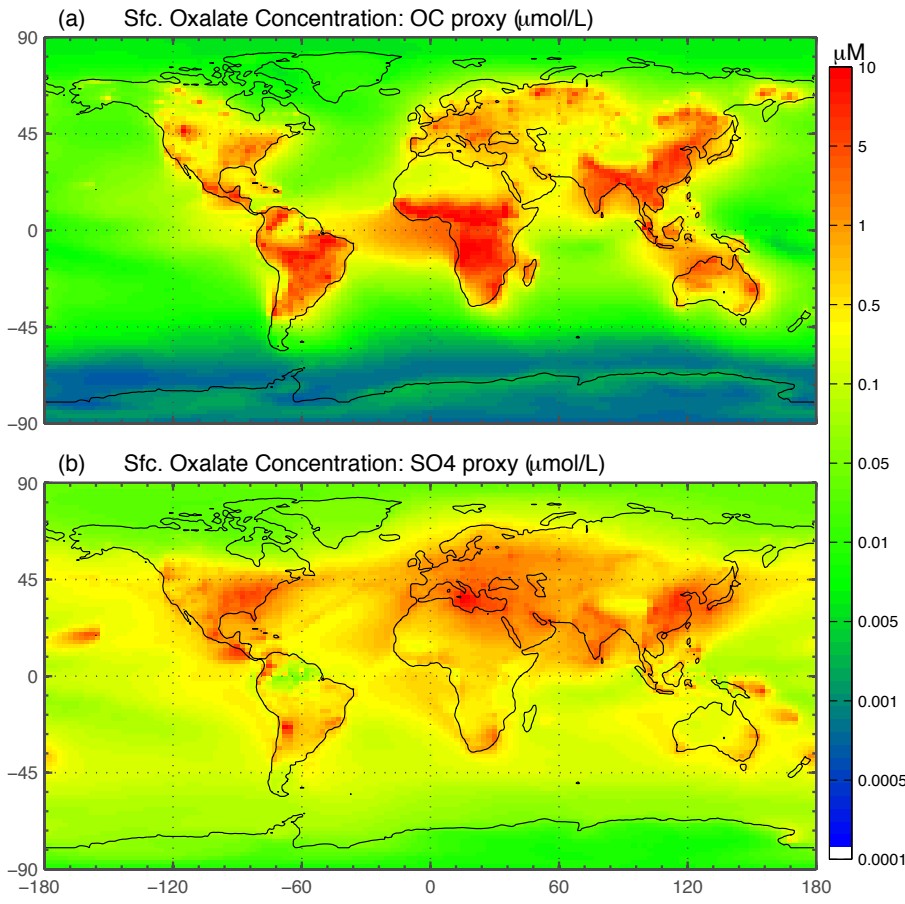





**Figure 3**

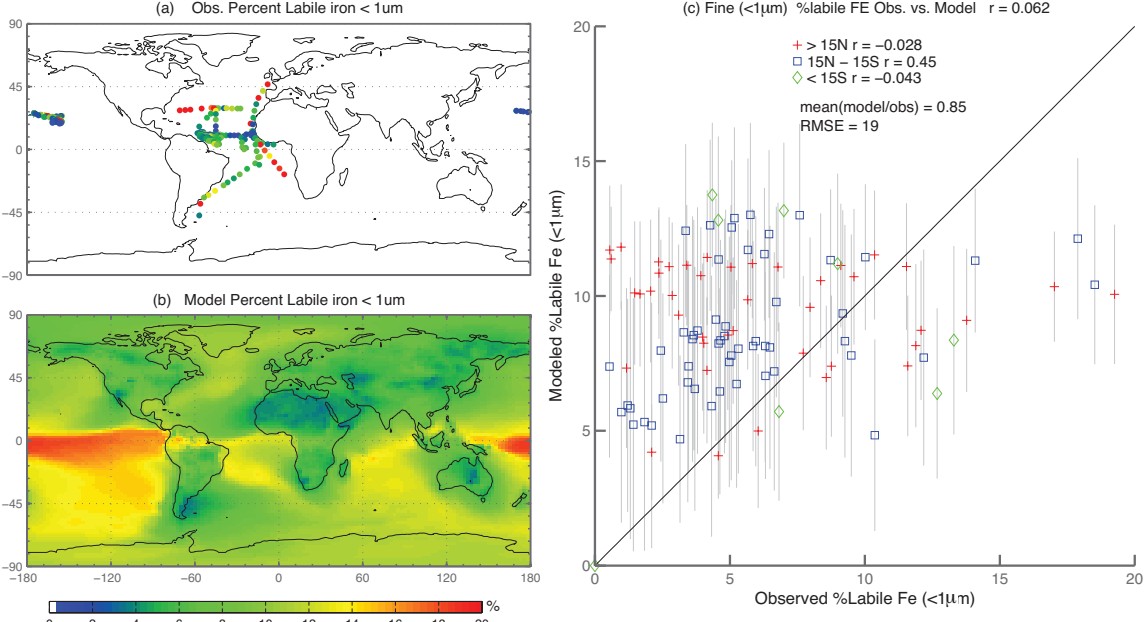




**Figure 4**

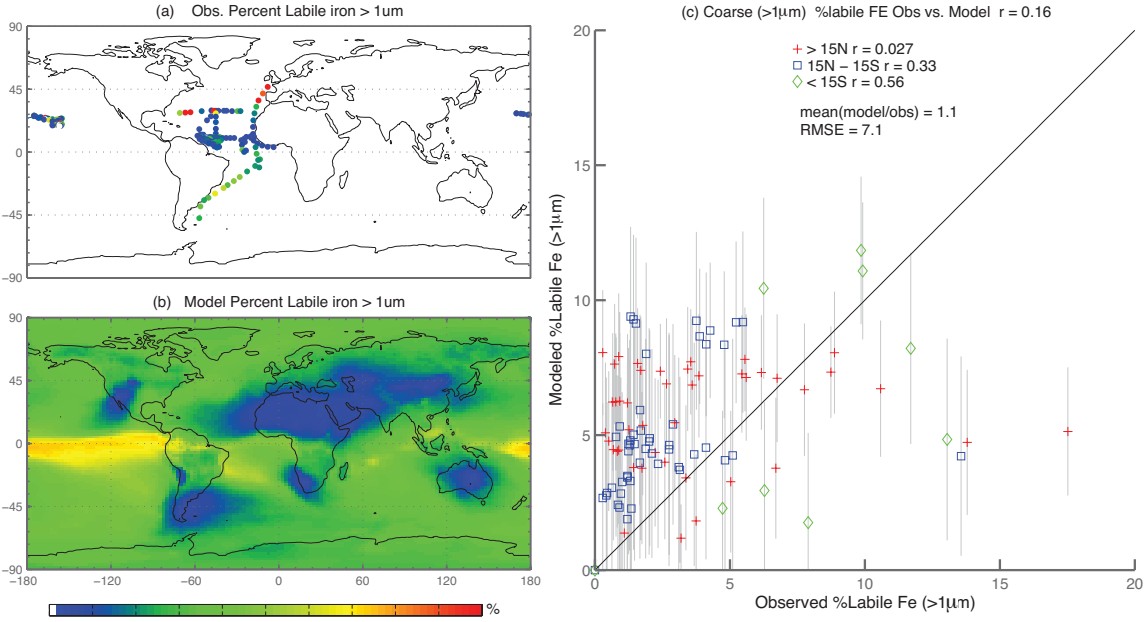

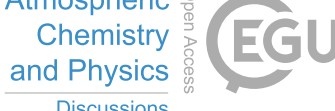

**Figure 5**

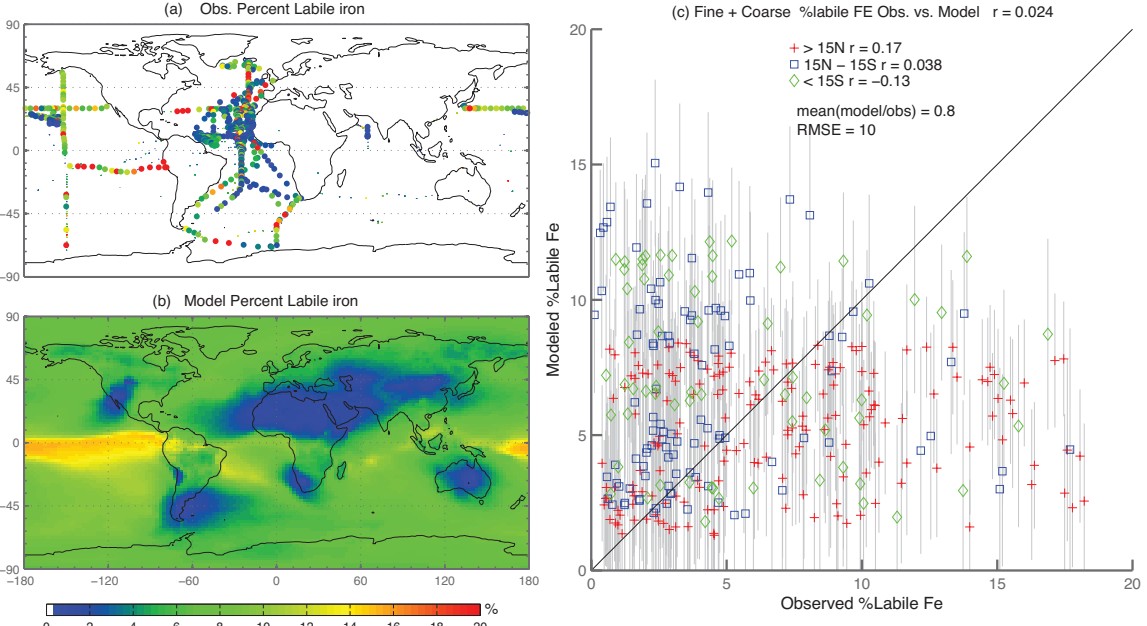





**Figure 6**

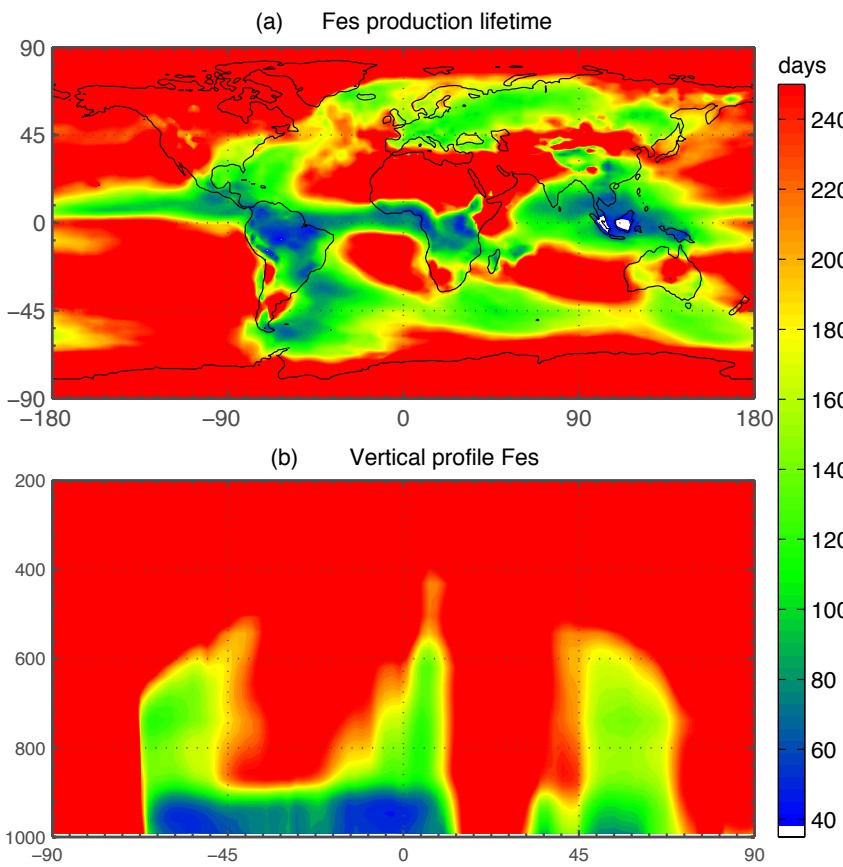





**Figure 7**

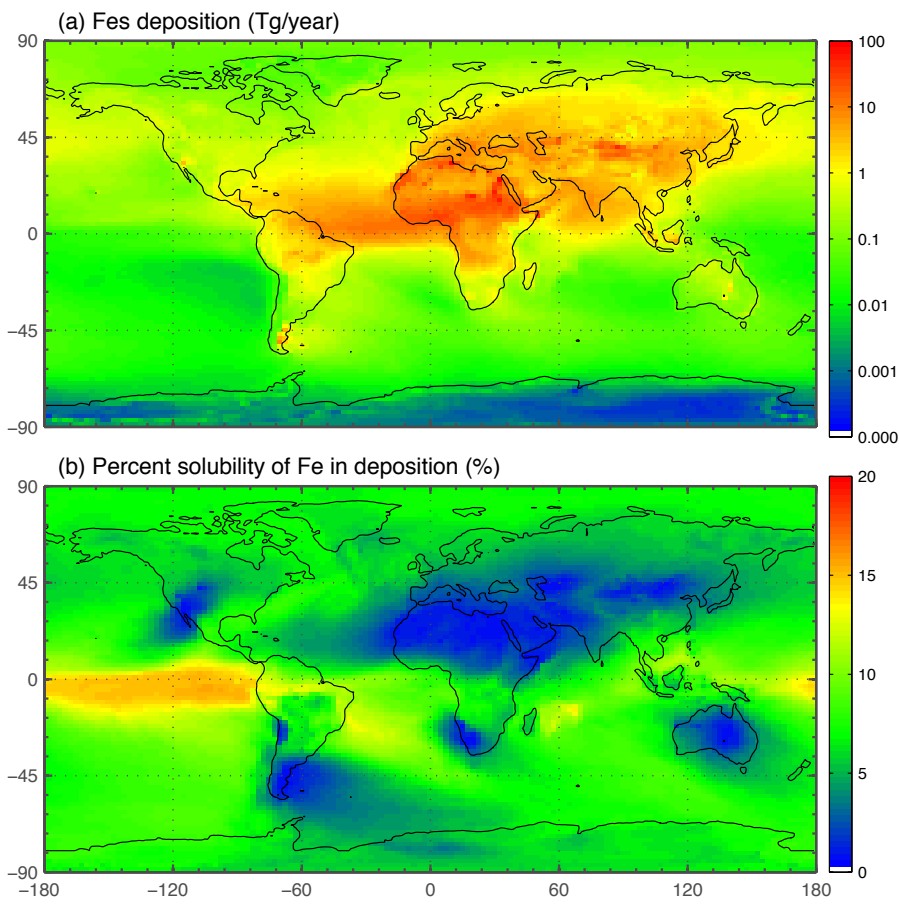



**Figure 8**

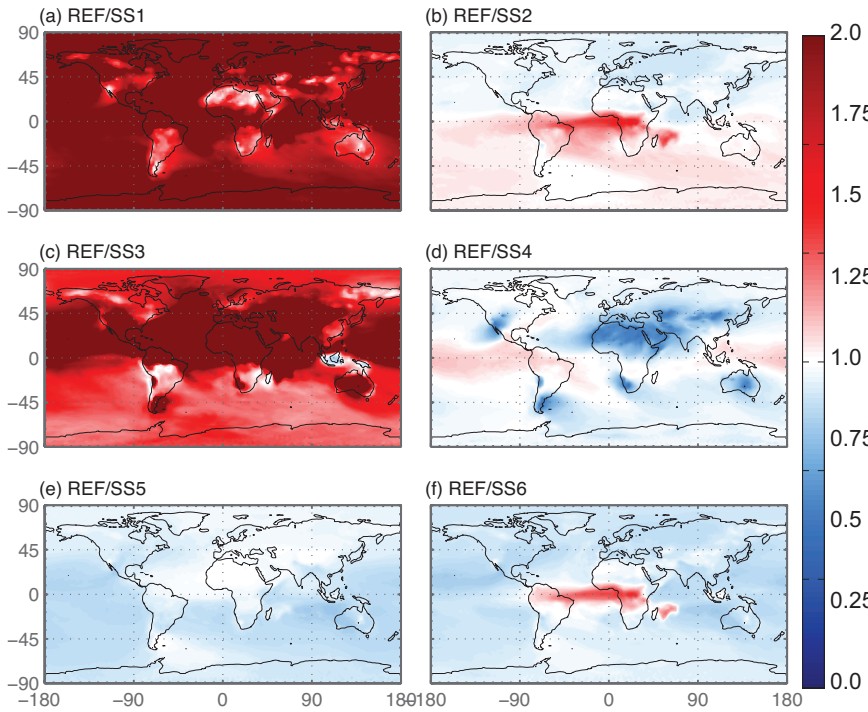


**Figure 9**

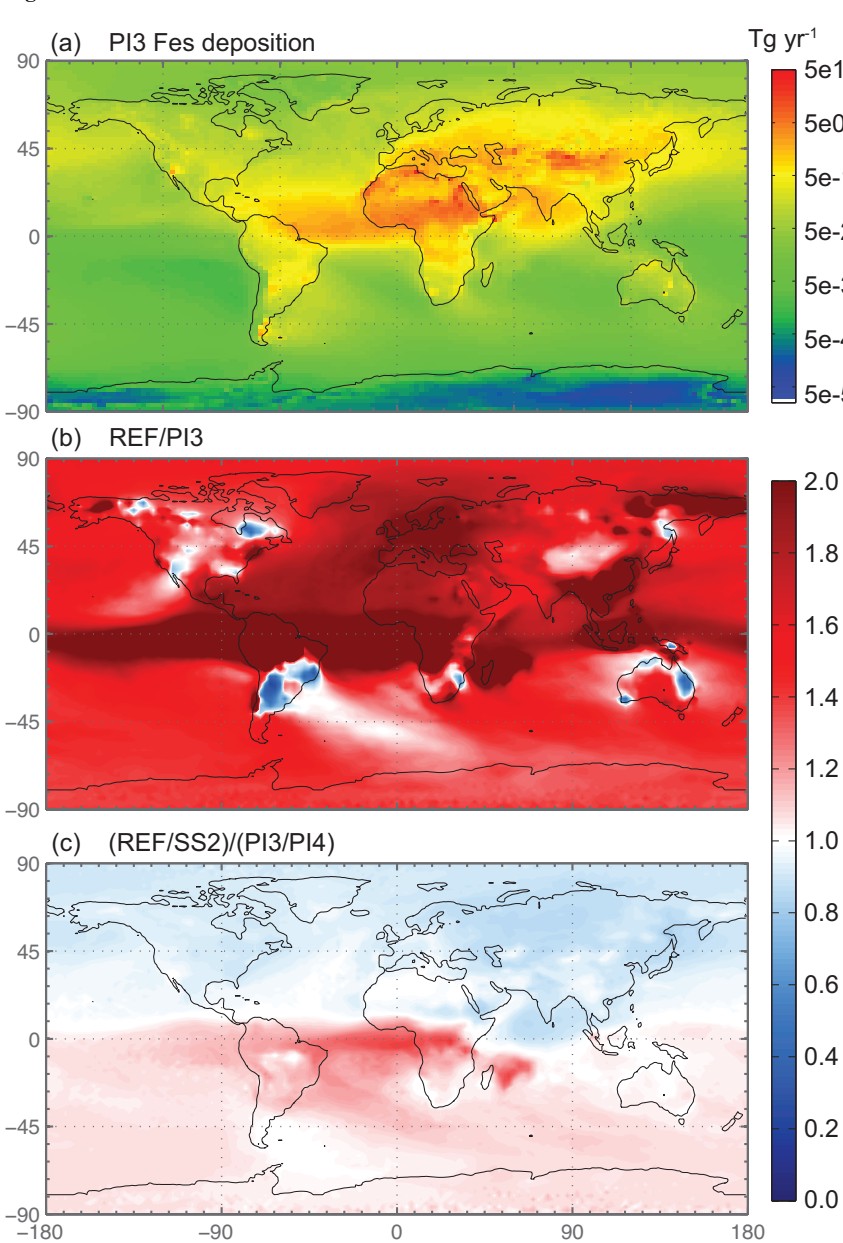



**Figure 10**

