# Peer review of "Atmospheric Processing of Iron in Mineral and Combustion Aerosols: Development of an Intermediate-Complexity Mechanism Suitable for Earth System Models."

_Atmospheric Chemistry and Physics, 2018_

## Referee Comment (RC1) · Anonymous Referee #1 · 22 Mar 2018

General comments

Parameterizations and simplifications in atmospheric processing of iron in aerosols are required for multi-decadal and centennial simulations. The authors presented a comprehensive modeling study of labile iron, assuming fast iron dissolution rates in cloudy grid boxes by applying "in-cloud" iron dissolution rates. The work conducted in this paper may contribute to improve our understanding of iron cycle, although more works will be required to improve the atmospheric processing of iron in aerosols. I have some comments and questions to improve this paper. In particular, there are many

tuning parameters which should be clarified for future studies, although the paper is well written.

Specific comments

Abstract

p.1., l.19, p.9, l.27, and p.16, l.25: Please rephrase "within range of the observational mean", as you explained "less than 1 signifies an underestimation" in p.11, l.32.

p.1., l.25, p.10, l.13, and Table 5a: This contradicts to the main conclusion from previous version (Luo et al., 2008). Figure S1 indicated the underestimates of iron below 15S, suggesting the omission of optimization of dust emissions. Since the total iron in combustion aerosols are emitted following Luo et al. (2008), you must predict lower dust emissions in the Southern Hemisphere. After you switched the coupled simulation to the off-line simulation, how did you scale the dust emissions? Since the dust emissions strongly depend on the meteorology, you needed to optimize the dust emissions. If you did not, please clarify the omission of optimization of dust emissions and the underestimates of iron below 15S.

1 Introduction

p.2, l.5: typo, HNLC.

p.3, l.13 and p.5, l.8: More recent study developed an iron dissolution scheme that reduced the number of mineral tracers for implementation in Earth system models (Ito and Shi, 2016).

2 Methods

p.4, l.15: Please clarify the link between three source modes and the size distribution across dust bins in p.4, l.18.

p.4. l.24: Please specify the sub-time step for multiple in-cloud cycles for aerosols and clarify the contact time of aerosols with cloud droplets.

[Figure]

p.6, l.25: Please show the results of pH from REF and SS5.

p.6, l.29 and Figure 1: Please correct the unit in Figure 1. The iron dissolution rates from Eq. (3) at oxalate concentration = 0 are higher than those from Eq. (1). However, the iron dissolved at oxalate concentration = 0 is already considered in FeRR. Please show the comparison of Fe solubility estimated using equations (1) and (3) from illite at oxalate concentration = 0 with Fig. S2 in Ito and Shi (2016). If you assume fast iron dissolution rates in cloudy grid boxes, please clarify it.

p.7, Eq. (2) and p.8, Eq. (4): How did you determine the scaling factor of 15?

p.7, l.8: Please show the comparison of oxalate from REF and SS2 with observations, assuming that oxalate remains in the particulate phase upon cloud evaporation. Please clarify if you assume high oxalate concentration in the aerosols in cloudy grid boxes.

p.8, l.14: Since these values strongly depend on the meteorology, you needed to optimize the tuning parameters (see the comments on p.8, l.18 and p.11, l.24). If you did not, please clarify the omission of optimization.

p.8, l.18: How did you determine the scaling factor of 5?

p.8, l.23: Guo et al. (2016) and Weber et al. (2016) excluded dust events and reported high acidity for fine particles. Please see Extended Data Figure 1 in Weber et al. (2016). The pH estimate is higher than 3 even though SO4 concentration is higher than Na concentration. I strongly recommend simulations with lower pH for combustion aerosols but higher pH for dust than REF in future studies.

p.8, l.28: How did you estimate preindustrial chemistry and dust sources assuming present day climate conditions?

3 Results

p.10, l.7: This is not the reason of the overestimates and underestimates (see the comments on p.1., l.25, p.10, l.13, and Table 5a).

p.11, l.5 and p.13, l.1: Why did you estimate higher iron solubility (i.e., overestimate) in the finest particle size bin off the coast of North Africa? The value is higher than the initial iron solubility for dust aerosols in Table S2d. However, the initial iron solubility for the finest particle size bin should be higher than that. Please show the emissions in finest particle size bin as in Table S2d.

p.11, l.21, Tables 3 and 7: What is tau?

p.11, l.24: Please indicate the values for "larger errors". The underestimates suggest the omission of optimization (see the comment on p.8, l.14).

p.13, l.30, Figure 8: Figures 8 (a) and (c) indicate the ratios of 2 over most regions. Did you set up the maximum ratios in your plots? Please correct it.

p.14: Please rephrase preindustrial "chemistry". This is confusing, because the chemistry depends on the meteorology.

Acknowledgments

p.17, l.16: D. H. is co-author.

Tables

p.21, Table 1: Please revise it to clarify the differences between REF and SS4.

References

Guo, H, et al. Fine particle pH and the partitioning of nitric acid during winter in the northeastern United States, J. Geophys. Res. Atmos., 121, 10355–10376, 2016.

Ito, A., and Shi Z.: Delivery of anthropogenic bioavailable iron from mineral dust and combustion aerosols to the ocean, Atmos. Chem. Phys., 16, 85–99, 2016.

Luo, C., Mahowald, N., Bond, T., Chuang, P. Y., Artaxo, P., Siefert, R., Chen, Y., and Schauer, J.: Combustion iron distribution and deposition, Global Biogeochem. Cy., 22, GB1012, 2008.

Weber, R. J., Guo, H., Russell, A. G., and Nenes, A.: High aerosol acidity despite declining atmospheric sulfate concentrations over the past 15 years, Nat. Geosci. 9, 282–285, 2016.

---

## Referee Comment (RC2) · Anonymous Referee #2 · 10 May 2018

This paper, develop an iron dissolution scheme of intermediate complexity that can be used in Earth system models. The overall presentation of the article is clearly structured, and the description of tests and calculations is also complete. However, there are some problems you should clearly explain and corrected before this paper is accepted. 1. Page 3, line 18, you said "We simulate four types of iron (readily-released Fe, medium-soluble Fe, slowsoluble Fe and combustion Fe)". But on page 5, line 24 , you have said "Three types of dust Fe are simulated in the model: readily-released iron (FeRR), medium-soluble iron (Femed) and slow-soluble or refractory iron (Feslow)".

[Figure]

There are some confusions, please explain clearly. 2. Page 6, line 13, it is recommended that "FeRR, Femed and Feslow" should be enclosed in parentheses. 3. In equation (1), R should be constant. However, there is no explanation in the following text, please explain clearly. 4. In equation (1), please explain how to calculate R. 5. In equation (2), please explain the significance of each physical quantity. 6. Page 7, line 15, you said "There tends to be more oxalate in the model simulations in tropical regions (Figure 2 from (Myriokefalitakis et al., 2011)) which is better captured in our model simulations using the OC+SOA versus the sulfate proxy for modeled oxalate concentrations.". Have you verified it? Please explain this. 7. Page 11, line 28, you have said "While the oxalate mechanism derived from the sulfate proxy (SS2) has marginally stronger correlations compared to the reference case, the difference between these is not statistically significant. ". Please explain the statistical relevance and variability mentioned. 8. Page 13, line 6, you said "Again, however, the fractional iron solubility is inversely related to total and soluble iron deposition, a result consistent with theory and observations". Please explain in detail the theories and observations mentioned in the text.

---

## Author Comment (AC1) · 30 Jun 2018

Referee comments are in black and the authors responses are in red. When text from the manuscript is quoted, new text is in bold face.

Referee 1: Interactive comment on "Atmospheric Processing of Iron in Mineral and Combustion Aerosols: Development of an Intermediate-Complexity Mechanism Suitable for Earth System Models" by Rachel A. Scanza et al.

General comments:

[Figure]

Parameterizations and simplifications in atmospheric processing of iron in aerosols are required for multi-decadal and centennial simulations. The authors presented a comprehensive modeling study of labile iron, assuming fast iron dissolution rates in cloudy grid boxes by applying "in-cloud" iron dissolution rates. The work conducted in this paper may contribute to improve our understanding of iron cycle, although more works will be required to improve the atmospheric processing of iron in aerosols. I have some comments and questions to improve this paper. In particular, there are many tuning parameters which should be clarified for future studies, although the paper is well written.

We thank the referee for the careful evaluation and helpful comments to improve the quality of the manuscript. We revise the text to clarify and address the referee's comments.

Specific comments: Abstract p.1., l.19, p.9, l.27, and p.16, l.25: Please rephrase "within range of the observational mean", as you explained "less than 1 signifies an underestimation" in p.11, l.32. We modify the text on page 1, lines 18-19: "We define a semi-quantitative metric as the model mean at points with observations divided by the observational mean (MMO). The model is in reasonable agreement with observations of fractional iron solubility with an MMO of 0.86."

We modify the text on page 9, line 24: "The MMO does not intend to evaluate the model's ability to capture observational variability but simply to assess if the model can reasonably estimate the observational mean."

We modify the text on page 16, line 25: "...fractional iron solubility MMO is 0.86, indicating that while the model is not capturing all of the observational variability, it is in reasonable agreement with the observational mean."

p.1.,l.25, p.10, l.13, and Table 5a: This contradicts to the main conclusion from previous version (Luo et al., 2008). Figure S1 indicated the underestimates of iron below 15S, suggesting the omission of optimization of dust emissions. Since the total iron

in combustion aerosols are emitted following Luo et al. (2008), you must predict lower dust emissions in the Southern Hemisphere. After you switched the coupled simulation to the off-line simulation, how did you scale the dust emissions? Since the dust emissions strongly depend on the meteorology, you needed to optimize dust emissions. If you did not, please clarify the omission of optimization of dust emissions and the underestimates of iron below 15S.

All model simulations in this study were conducted with optimized dust emissions following Albani et al., 2014 and forced with MERRA reanalysis meteorology. In Albani et al, 2014, the dust is optimized to best match AOD in the source regions, deposition and concentration. As in other models (Hunees et al., 2011), we cannot match both deposition and surface concentrations at the same time. Here we only compare to surface concentrations and add the following test on page 4 to make this more clear: "The dust module was tuned to best match aerosol optical depth (AOD), deposition and surface concentration data (Albani et al., 2014). Similar to other models (Huneeus et al., 2011), our model is unable to simultaneously match the surface concentration and deposition data in remote regions, and since here we only show concentration data, it will appear that the model overpredicts surface concentrations of dust."

TableS2d lists emissions of total iron in dust and combustion aerosols at 56.9 and 1.9 Tg/year respectively, and is very close to the total iron emissions reported in Luo et al., 2008 (55 and 1.7 Tg/year). The main conclusion in Luo et al., 2008 is "we obtain the result that deposition of soluble iron from combustion contributes 20 – 100% of the soluble iron deposition over many ocean regions." Therefore, our conclusions support those of that study and are within their range. The model we use in this study is structurally different to the MATCH model used in Luo et al., 2008 and the reanalysis meteorology used to drive MATCH is NCEP, compared to MERRA reanalysis meteorology used here; therefore, a like-for-like comparison of results with those in Luo et al., 2008 cannot be done as the Reviewer suggests and differences are instead a reflection of the different model structures and meteorology data sets used.

[Figure]

We also add the following text on page 10, lines 16-17: "...surface concentrations of dust are generally over-predicted with the exception of remote SH ocean regions. In addition, most dust models exhibit a low bias in SH dust deposition (e.g., Huneeus et al., 2010;Huneeus et al., 2011;Albani et al., 2014)."

Introduction p.2, l.5: typo, HNLC.

fixed.

p.3, l.13 and p.5, 1.8: More recent study developed an iron dissolution scheme that reduced the number of mineral tracers for implementation in Earth system models (Ito and Shi, 2016).

We modify the text by adding the following sentence on page 3, lines 9-10: "A more recent study developed an iron dissolution scheme with fewer mineral tracers to allow for simulations using Earth system models (Ito and Shi, 2016)."

Methods p.4, l.15: Please clarify the link between three source modes and the size distribution across dust bins in p.4, l.18 The particle size distributions for each dust bin are calculated from the mass fraction from the source trimodal PDF to the transport bins following (Eq. 12, Zender et al., 2003). The dust bins are bounded at 0.1, 1.0, 2.5, 5.0 and 10.0um with prescribed mass median diameter and standard deviation of 3.5um and 2.0, respectively. Three source modes described in Shulz et al., 1998, D'Almeida, 1987 and Zender et al., 2003 are defined by observed mass median diameter and geometric standard deviation (see Table 1, Zender et al., 2003). The portion of the mass fraction from each source mode to each of the 4 dust bins is described in (Table 2, Zender et al., 2003), and the bin fractions are modified following brittle fragmentation theory (Kok, 2011).

We modify the text by adding the following on page 4, lines 16-20:

"The three source modes are defined by observed mass median diameter and geometric standard deviation (d'Almeida, 1987;Schulz et al., 1998;Table 1, Zender et al.,

2003). Bin widths are prescribed at 0.1-1.0, 1.0-2.5, 2.5-5.0, and 5.0-10.0 $\mu$m (Mahowald et al., 2006;Zender et al., 2003) and have fixed lognormal sub-bin distributions (mass median diameter = 3.5$\mu$m, $\sigma$g = 2.0) (Zender et al., 2003). The size distribution across dust bins was modified from the release version of the model to follow the brittle fragmentation theory of vertical dust flux . . ."

p.4, l.24: Please specify the sub-time step for multiple in-cloud cycles for aerosols and clarify the contact time of aerosols with cloud droplets. CAM4 aerosols do not interact with clouds with the exception of modeled iron and sulfate particles, the chemistry of which is explicitly simulated within clouds using the simple approximations developed here. In this study, with the CAM4 aerosol configuration, we are not actually simulating dissolution and evaporation of dust and oxalate particles within a cloud droplet but rather estimating it via simplified dissolution rates (Eq. 1-3).

We modify the text on page 4, lines 25-28: "CAM4 allows for multiple cycles of condensation and evaporation (Gent et al., 2011;Hurell et al., 2013;Neale et al., 2013) in order to match observational estimate of approximately three in-cloud cycles for aerosols (Lelieveld et al., 1998;Crutzen and Zimmerman, 1991); the model time step is 30 minutes with 20 sub-time steps for in-cloud chemistry."

p.6, l.25: Please show the results of pH from REF and SS5. Added a new figure for Supplement (Figure S3).

p.6, l.29 and Figure 1: Please correct the unit in Figure 1. The iron dissolution rates from Eq. (3) at oxalate concentration = 0 are higher than those from Eq. (1). However, the iron dissolved at oxalate concentration = 0 is already considered in FeRR. Please show the comparison of Fe solubility estimated using equations (1) and (3) from illite at oxalate concentration = 0 with Fig. S2 in Ito and Shi (2016). If you assume fast iron dissolution rates in cloudy grid boxes, please clarify it.

Units in figure one RFe vs. pH:

$RFe = K(T) \times a(H)\hat{}m \times f(\text{âĹĞG}) \times A \times MW$ where,

$K(T)$ has units of (mol m-2 s-1) $a(H)\hat{}m$ is unitless $f(\text{âĹĞG})$ is unitless $A$ has units of (m2 g-1) $MW$ has units of (g mol-1) âĽť "mol" /("m" ˆ2 "s" )×"m" ˆ2/"g" ×"g" /"mol" = s-1

The iron dissolution rate at [C2O42-] = 0 utilizes Table 4 in Paris et al., with our method for best fit which was a result of multiple different simulations where the intercept was examined to find the best match observations of %Fes. Regardless of the presence of C2O42-, the non-zero intercept with units of (s-1) for Ki,oxalate accounts for (non-oxalate) in-cloud organic ligand processing which is always present.

We add the following text on page 7: "The iron dissolution rate at [C2O42-] = 0 utilizes (Table 4, Paris et al., 2011) with our method for best fit which was a result of multiple different simulations where the intercept was examined to find the best match to observations of %Fes. The non-zero intercept for Ki,oxalate accounts for (non-oxalate) in-cloud organic ligand processing."

We assume iron dissolution rate in cloudy grid boxes following the higher solubility for Femed, which can be found in the text on page 6, line 33 "…we use the Femed dissolution rate for the remaining combustion iron.

We also add a figure in the supplement, Figure S4 showing the Fe solubility from illite with time.

p.7, Eq. (2) and p.8, Eq. (4): How did you determine the scaling factor of 15? Multiple simulations were performed to choose our reference case and the factor of 15 produced the best match to previous modeling studies with more explicitly simulated oxalate. Since CAM4 aerosols do not interact with clouds, it was necessary to parameterize this interaction and to select an appropriate scaling factor to best match the oxalate distribution (Fig. 2, Myriokefalitakes et al., 2011) and observations of fractional iron solubility.

We added the following text on page 7, lines 10-13: "In choosing our reference case,

multiple simulations were conducted to best match observations of fractional iron solubility and the surface distribution of oxalate (Figure 2a,c, Myriokefalitakis et al., 2011); the factor of 15 in Eq. (2) yielded the best results."

p.7, l.8: Please show the comparison of oxalate from REF and SS2 with observations, assuming that oxalate remains in the particulate phase upon cloud evaporation. Please clarify if you assume high oxalate concentration in the aerosols in cloudy grid boxes. The oxalate concentration in cloudy gridboxes is calculated with Equation 2. Regions with both clouds and oxalate concentrations shown in Figure 2 will have high oxalate concentrations and thus faster dissolution. We are unable to show a direct comparison of REF and SS2 particulate phase oxalate given that CAM4 aerosols (except sulfate) do not interact with clouds and were only parameterized using proxy species. However, we refer the Reviewer to Figure 2 in this study and Figure 2 (a,c) in Myriokefalitakis et al., 2011, which shows good agreement when using organic carbon as the proxy for oxalate in CAM4. This has been stated in the manuscript already so no additional text is required.

p.8, l.14: Since these values strongly depend on the meteorology, you needed to optimize the tuning parameters (see comments on p.8, l.18 and p.11, l.24). If you did not, please clarify the omissions of optimization. Tuning parameters for dust emission were optimized following Albani et al., 2014. Page 8, lines 14-16 describe the very simple iron dissolution parameterization for SS3 from Hand et al., 2004. Please see our response to the second specific comment.

p.8, 1.18: How did you determine the scaling factor of 5? Simulations in Hand et al., 2004 did not take into account any combustion iron. Combustion iron vs dust iron solubility is not well understood. Particularly, a wide range of initial dust solubility (0.1 – 1+% (Luo et al., 2008;Jickells et al. 2005;Shi et al., 2012;Johnson and Meskhidze, 2013;Ito and Xu, 2014). Using our reference case, the annually-average global Fes emissions (0.5069 Tg/year) divided by total dust Fe emissions (56.9 Tg/year) yields an initial dust %Fes of 0.89%. Assuming combustion %Fes of 4%, this corresponds to an

enhancement of 4.5 times (although this could potentially range from 40 to less than 4 times).

We add the following text on page 8, lines 19-24: "Hand et al., 2004 did not take into account iron from combustion aerosols and there are a wide range of reported dust Fe solubilities at emission, 0.1-1+% (Luo et al., 2008;Jickells et al. 2005;Shi et al., 2012;Johnson and Meskhidze, 2013;Ito and Xu, 2014). To account for the typically higher solubilities associated with combustion iron, and using Fes/Fet for dust iron (Table S2d), we assume an initial solubility for combustion and dust Fe of 4% and 0.89%, respectively; this corresponds to an enhancement of 4.5 times and hence chose, somewhat arbitrarily, to increase the dissolution rate by a factor of 5."

p.8, l.23: Guo et al., 2016 and Weber et al., 2016 excluded dust events and reported high acidity for fine particles. Please see Extended Data Figure 1 in Weber et al., 2016. The pH estimate is higher than 3 even though SO4 concentration is higher than Na concentration. I strongly recommend simulations with lower pH for combustion aerosols but higher pH for dust than REF in future studies. This is an excellent suggestion and will be implemented in our current model development using this iron dissolution scheme within the modal aerosol module (MAM) in the CESM (Hamilton et al., in prep).

p.8, l.28: How did you estimate preindustrial chemistry and dust sources assuming present day climate conditions? Preindustrial chemistry is estimated from historical emission from CMIP5 and include reduced emissions of sulfate, OC and BC. (Lamarque et al., 2010). Preindustrial dust sources from 1850 are estimated based on those used in Mahowald et al., 2006 for late 19th century. Present day climate conditions indicate that we use the current MERRA reanalysis meteorology (e.g., temperature, wind speed, etc.), but should not be confused with chemistry which refers only to the atmospheric compositional state (in this case CO2 and oxidant concentrations. We add the following text on page 9: "Here we refer to "preindustrial chemistry" as historical emissions for CMIP5 (Lamarque et al., 2010) and includes reduced emissions for

sulphate, OC, and BC. "

3 Results p.10, l,7: This is not the reason of the overestimates and underestimates (see the comments on p.1., l.25, p.10, l.13 and Table 5a). Dust emissions are optimized in all simulations; please see our previous response.

p.11, l.5 and p.13, l.1: Why did you estimate higher iron solubility (i.e., overestimate) in the finest particle size bin off the coast of North Africa? The value is higher than the initial iron solubility for dust aerosols in Table S2d. However, the initial iron solubility for the finest particle size bin should be higher than that. Please show the emissions in the finest particle size bin as in Table S2d.

The initial iron solubility for dust aerosols from Table S2d is 0.89% and when we include combustion aerosols the combined initial iron solubility for all bins is 0.99% (4% Fecomb). Fine mode emissions in units of kg m-2 s-1 for dust, Fetdust, Fetcomb, Fesdust, Fescomb are 1.93E+10, 1.02E+09, 2.4E+08, 3.6E+07 and 9.7E+06 respectively. The readily released iron from Ito and Xu, 2014 is presumed already soluble in bin1 and for bins 2-4, FeRR from kaolinite and feldspar is prescribed as soluble while 25% of FeRR from illite and smectite is prescribed, as stated in Section 2.3, lines 24-33. This corresponds to a higher intital solubility of dust iron in the finest size bin (3.6%); including combustion yields 3.7%.

We modify the Supplementary Material by adding a column in Table S2d for the fine mode emissions.

p.11, l.21, Tables 3 and 7: What is tau? In tables 3 and 7, tau is the annually-averaged global mean of insoluble iron turnover time (days). It is defined as the total insoluble iron from dust and combustion aerosols divided by the production of soluble iron from insoluble iron.

We add the following text to the table caption for Table 3 (page : "In the last two rows, tau is the annually-averaged global mean insoluble iron turnover time (days) and is

defined as the total insoluble iron from dust and combustion aerosols divided by the production of soluble iron from insoluble iron. Mean %fesdep is the average fractional iron solubility at deposition to global ocean basins."

We add the following text to the caption for Table 7: "...percent difference for the average production lifetime of Fes (days) labeled here as "tau" and the..."

p.11, l.24: Please indicate the values for "larger errors." The underestimates suggest the omission of optimization (see the comment on p.8, l.14). Dust emissions are optimized in all simulations; please see our previous response.

p.13, l.30, Figure 8: Figures 8 (a) and (c) indicate the ratios of 2 over most regions. Did you set up the maximum ratios in your plots? Please correct it.

Yes we did and have modified the figure to account for this.

p.14: Please rephrase preindustrial "chemistry". This is confusing, because the chemistry depends on the meteorology.

Preindustrial chemistry is estimated from CMIP5 historical emissions (Lamarque et al. , 2010), which should not be confused with the MERRA reanalysis meteorology used to drive all simulations in this study.

Acknowledgments p.17, l.16: D. H. is co-author fixed.

Tables p.21, Table 1: Please revise it to clarify the differences between REF and SS4.

SS4 is REF with spatial dependence of iron on mineralogy removed and the global average iron and calcite concentrations from the reference case are prescribed at emission. We add the following text in caption for Table 1a to clarify this difference: "SS4 tracers have no spatial dependence on mineralogy and have prescribed global average fractions from REF at emission."

---

## Author Comment (AC2) · 30 Jun 2018

Referee comments are in black and the authors responses are in red. When text from the manuscript is quoted, new text is in bold face.

Referee 2: Interactive comment on "Atmospheric Processing of Iron in Mineral and Combustion Aerosols: Development of an Intermediate-Complexity Mechanism Suitable for Earth System Models" by Rachel A. Scanza et al.

This paper, develop an iron dissolution scheme of intermediate complexity that can

be used in Earth system models. The overall presentation of the article is clearly structured, and the description of tests and calculations is also complete. However, there are some problems you should clearly explain and corrected before this paper is accepted.

We thank the referee for the careful evaluation and helpful comments to improve the quality of the manuscript. We revise the text to clarify and address the referee's comments.

Page 3, line 18, you said, "We simulate four types of iron (readily-released Fe, medium soluble Fe, slow soluble Fe and combustion Fe)". But on page 5, line 24, you have said "Three types of dust Fe are simulated in the model: readily-released iron (FeRR), medium-soluble iron (Femed) and slow-soluble or refractory iron (Feslow)". There are some confusions, please explain clearly. We account for three types of iron from dust and one type from combustion. We modify the text on page 3, line 17-18: "We simulate four types of iron, three for mineral dust (readily-released Fe, medium soluble Fe and slow-soluble Fe) and one for combustion aerosols (combustion Fe). "

2. Page 6, line 13, it is recommended that "FeRR, Femed and Feslow" should be enclosed in parentheses.

We modify the text on page 6: "...three types of iron in dust (FeRR, Femed, and Feslow)."

3. In equation (1), R should be constant. However there is no explanation in the following text, please explain clearly. In Eq. (1), R is a function of temperature. RFe=K(T)×a(H)ˆm×f(âĹĞG)×A×MW where,

K(T) is a function of temperature and has units of (mol m-2 s-1)

We add the following text on page 6: "where RFe_i is a function of temperature and has units (s-1), i represents either medium or slow soluble Fe, K_i (T) in units of (moles m-2 s-1) is the temperature dependent rate coefficient (Table 8, Meskhidze et al., 2005),

a(Hˆ+) is the proton concentration with an empirical reaction order mi, f(âĹĞG_r) accounts for the change in the dissolution rate with variation from equilibrium (and equals 1 for simplicity (Luo et al., 2008)), A_i is the specific surface area of minerali in units of (m2 g-1) and MW_i is the molecular weight in units of (g mol-1) for minerali."

4. In equation (1), please explain how to calculate R.

RFe=K(T)×a(H)ˆm×f(âĹĞG)×A×MW For example, K(T) utilizes Tables 8 in Meskidze et al., 2005 for illite and hematite (stage II) (Ito and Xu, 2014), a(H+)m is the pH, f(âĹĞG) is set to 1 following Luo et al., 2008, A is the specific surface area of illite (Femed), hematite (Feslow) and MW is simply the molecular weight of either illite or hematite, depending on which dissolution rate you are calculating.

5. In equation (2), please explain the significance of each physical quantity.

In Eq. (2), [C2O42-] is the concentration of oxalate in each atmospheric gridbox. [OC] and [SOA] are the organic carbon and secondary organic aerosol concentrations (mass mixing ratio) in each gridbox, and these species are included in the release version of CESM. The denominator is the maximum of the sum of global averaged mass mixing ratios for OC & SOA, to ensure that the highest possible oxalate concentration is 15 $\mu$mols L-1, this factor was chosen to best match observations of fractional iron solubility and the spatial distribution of oxalate from Myriokefalitakis et al., 2011. We've added the following text on page 7, lines 10-13: "In choosing our reference case, multiple simulations were conducted to best match observations of fractional iron solubility and the surface distribution of oxalate (Figure 2a,c, Myriokefalitakis et al., 2011); the factor of 15 in Eq. (2) yielded the best results."

6. Page 7, line 15, you said "There tends to be more oxalate in the model simulations in tropical regions (Figure 2 from Myriokefalitakis et al., 2011) which is better captured in our model simulations using the OC+SOA versus the sulfate proxy for modeled oxalate concentrations. Have you verified it? Please explain this.

Visually comparing Figure 2 in this study with Figure 2 (a) and (c) in Myriokefalitakis et al., it should be clear that the spatial distribution of oxalate using the OC+SOA (a) vs the sulfate (b) proxy is a better match to Myriokefalitakis et al., 2011.

7. Page 11, line 28, you have said "While the oxalate mechanism derived from the sulfate proxy (SS2) have marginally stronger correlations compared to the reference casem the difference between these is not statistically significant". Please explain the statistical relevance and variability mentioned.

We remove this sentence as the correlation coefficients for all cases are poor.

8. Page 13, line 6, you said, "Again, however, the fractiona; iron solubility is inversely related to total and soluble iron deposition, a result consistent with theory and observations". Please explain in detail the theories and observations mentioned in the text.

Many studies assume that OPP requires bioavailable iron and report increased iron liberation from ferric oxides with decreasing pH acidic species (Duce and Tindale, 1991;Zhu et al., 1997;Zhuang et al., 1992;Jickells and Spokes, 2001;Desboeufs et al., 2001;Meskhidze et al., 2003). (Chen and Siefert, 2004) and (Baker and Jickells, 2006) find decreasing fractional iron solubility with mineral dust concentrations for different atmospheric environments, concluding that solubility is partially a function of particle size (Baker and Croot, 2010). Smaller particles have longer atmospheric lifetimes and thus higher probabilities of undergoing chemical reduction. Additionally, combustion aerosols whose fractional iron solubility is higher are generally small particles Particle size has been implicated in the bioavailable iron problem with many studies finding that smaller particles with a larger surface area to volume ratio have increased dissolution (Baker and Jickells, 2006). This observation could result because smaller particles dominate long-range transport as larger particles are preferentially removed due to gravity. For example, (Hand et al., 2004) finds that soluble iron is several times greater in the fine dust mode than the coarse mode without including a surface area effect. More recently however, iron solubility increasing with decreasing particle size

has been disputed (Shi et al., 2012;Buck et al., 2010), indicating that particle size can only partially account for the differences in solubility. The implication that particle size enhances solubility does not take into account that the mineralogy of the dust distribution changes during transport, with clay enrichment due to the larger particles sizes of quartz and feldspar (Glaccum and Prospero, 1980). The mineralogy of dust is important with different minerals contributing different forms of iron (i.e. iron oxides, nanoparticle coating, alluminosilicate substitutions) (Journet et al., 2008;Ito and Xu, 2014;Meskhidze et al., 2005). Observations from Sholkovitz et al., 2012 show a trend of increasing iron solubility with decreasing total iron concentration and conclude that much of this is due to the smaller particle size distribution of combustion aerosols and their ability to enhance dust iron dissolution. Mahowald et al, in press, show using a simple 1-d plume model, that either differential solubility in emissions of combustion aerosols, or atmospheric processing of dust iron can match the observed relationship described above. At emission, combustion aerosols are typically smaller and thus remain suspended in the atmosphere longer than the larger dust aerosols. In addition, since combustion aerosols appear to have higher Fe solubility, the plume will increase in solubility as it is transported downwind from continental sources. Alternatively, if only dust aerosols are considered to be sources of iron, but are atmospherically processed downwind from source regions, one will see lower values of iron in the aerosols, as particles fall out yet the remaining iron will be more soluble (longer atmospheric processing). Quantitative comparison of this simple plume model indicates that either of these two cases can match observations (Mahowald et al., in press).

We include the following text on page 2: "Observations from Sholkovitz et al., 2012 show a trend of increasing iron solubility with decreasing total iron concentration and conclude that much of this is due to the smaller particle size distribution of combustion aerosols and their ability to enhance dust iron dissolution. Mahowald et al, in press, show using a simple 1-d plume model, that either differential solubility in emissions of combustion aerosols, or atmospheric processing of dust iron can match the observed relationship described above. At emission, combustion aerosols are typically smaller

and thus remain suspended in the atmosphere longer than the larger dust aerosols. In addition, since combustion aerosols appear to have higher Fe solubility, the plume will increase in solubility as it is transported downwind from continental sources. Alternatively, if only dust aerosols are considered to be sources of iron, but are atmospherically processed downwind from source regions, one will see lower values of iron in the aerosols, as particles fall out yet the remaining iron will be more soluble (longer atmospheric processing). Quantitative comparison of this simple plume model indicates that either of these two cases can match observations (Mahowald et al., in press). "
* * *

---

## Author Response (AR2)

Author comments are in italic font while the authors response is in standard font. When text from the manuscript is quoted, new text is in **bold face**.

**5 General comments**

The paper has been improved significantly. I can recommend it for publication in ACP after the authors respond to my comments.

10 Specific comments

Abstract

p.2, 1.32: Mahowald et al. (2018) did not show the Sholkovitz plot, but showed log graphs. Please show it with a linear graph. The combustion case (blue case) only can reproduce the high solubility at low concentration.

15

For very high solubility, the combustion case is the only case that can match these; however, for most of the observed data, including solubilities up to 30%, both the atmospheric processing and combustion aerosols cases in the 1-D plume model (Mahowald et al., 2018) can reasonably match the inverse trend from Sholkovitz et al., 2012.

20 We add the following text on page 3, line 8: "Panel (b) in Box 2 (Mahowald et al., 2018) is reproduced here (Figure S5) on a linear scale to facilitate a comparison to Figure 3 in Sholkovitz et al., 2012). While the combustion case (blue) is the only case able to replicate very high aerosol Fe solubility (>30%), all three cases match the observed trend of increasing fractional iron solubility with decreasing total aerosol iron concentrations."

25 Methods

p5., l.13: Please indicate the graph of the model over-predictions of surface concentration of dust.

We include a reference to the graph of dust surface concentration on page 5, line 15: "...regions, (Figure 3 in Albani et al., 2014), and since..."

1

**Atmospheric Processing of Iron in Mineral and Combustion Aerosols: Development of an Intermediate-Complexity Mechanism Suitable for Earth System Models.**

Rachel A. Scanza1,2, Douglas S. Hamilton1, Carlos Perez Garcia-Pando3, Clifton Buck4, Alex Baker5, Natalie M. Mahowald1

[revised manuscript text omitted]
- Okin, G. S.: A new model of wind erosion in the presence of vegetation, Journal of Geophysical Research: Earth Surface, 113, 2008.
   Okin, G. S., Baker, A. R., Tegen, I., Mahowald, N. M., Dentener, F. J., Duce, R. A., Galloway, J. N., Hunter, K., Kanakidou, M.,
- 45 Okin, G. S., Baker, A. R., Tegen, I., Mahowald, N. M., Dentener, F. J., Duce, R. A., Galloway, J. N., Hunter, K., Kanakidou, M., and Kubilay, N.: Impacts of atmospheric nutrient deposition on marine productivity: Roles of nitrogen, phosphorus, and iron, Global Biogeochemical Cycles, 25, 2011. 

[revised manuscript text omitted]

-